# Phase Transition from Clean Training to Adversarial Training

**Yue Xing**
Department of Statistics
Purdue University
xing49@purdue.edu

**Qifan Song**
Department of Statistics
Purdue University
qfsong@purdue.edu

**Guang Cheng**
Department of Statistics
University of California, Los Angeles
guangcheng@ucla.edu

## Abstract

Adversarial training is one important algorithm to achieve robust machine learning models. However, numerous empirical results show a great performance degradation from clean training to adversarial training (e.g., 90+% vs 67% testing accuracy on CIFAR-10 dataset), which does not match the theoretical guarantee delivered by the existing studies. Such a gap inspires us to explore the existence of an **(asymptotic)** phase transition phenomenon with respect to the attack strength: adversarial training is as well behaved as clean training in the small-attack regime, but there is a sharp transition from clean training to adversarial training in the large-attack regime. We validate this conjecture in linear regression models, and conduct comprehensive experiments in deep neural networks.

## 1 Introduction

Among various algorithms towards adversarially robust models, adversarial training is a popular and simple way to improve the adversarial robustness of the model. Many studies improve adversarial training from either theoretical or empirical aspects.

To justify the theoretical properties, an abundant literature studies the performance of adversarially robust models under common statistical models, e.g., linear regression, Gaussian mixture model and etc. in Javanmard et al. (2020); Mehrabi et al. (2021); Javanmard and Soltanolkotabi (2020); Dan et al. (2020); Taheri et al. (2020); Xing et al. (2021c,a). Besides, some other studies work on deep learning models. For example, Gao et al. (2019); Zhang et al. (2020b); Allen-Zhu and Li (2020) show the convergence of adversarial training in neural networks when the attack strength is sufficiently small. Other literature, e.g., Zhang et al. (2019); Wang et al. (2019b,a), propose improvements in the training loss of adversarial training.

However, there are some gaps between the promising theoretical results and the actual performance of adversarial training in real practice. First, the aforementioned theoretical works consider either simple models or sufficiently small attack strength. For simple models, adversarial training is well-behaved because the adversarial loss is still convex. For complicated models with small attacks, they conduct a similar analysis for adversarial training as for clean training, in the sense that adversarial training only introduces some additional diminishing error. Such settings are different from the setups used in real practice. Second, some other studies also reveal several potential concerns of adversarial training in deep learning, e.g., the smoothness problem Lee and Chandrakasan (2020); Xie et al. (2020); Xing et al. (2021a), overfitting issue Rice et al. (2020); Wu et al. (2020). Based on RobustBench (https://robustbench.github.io/), the best robust accuracy for CIFAR-10 using state-of-the-art methods is 66.6% Rebuffi et al. (2021), much worse than the clean accuracy of above 90% in clean training.

36th Conference on Neural Information Processing Systems (NeurIPS 2022).

As our attempt to fill these gaps, this work focuses on the role of adversarial attack strength. For instance, we are wondering whether taking 8/255 $\mathcal{L}_\infty$ attack is too strong for CIFAR-10 adversarial training using the current state-of-the-art neural networks architectures and optimizer configurations. Although clean training generalizes well in this setup and adversarial training can be viewed as an adaptation of clean training, they may act differently.

This paper investigates how adversarial training deviates from clean training as the attack strength grows. An illustration of this can be found in Figure 1. Under small attack strength $\epsilon$, the trajectory of adversarial training is close to that of clean training so that they converge to the same flat minima. As $\epsilon$ increases, the adversarial training process deviates from the clean training dramatically. Our aim is to determine a critical attack level, denoted as $\epsilon^*$, such that (i) when $\epsilon \ll \epsilon^*$, (S)GD trajectories of clean training and adversarial training are similar in some sense so that they can be handled similarly; but (ii) when $\epsilon \gg \epsilon^*$, the adversarial training will significantly deviate from clean training.

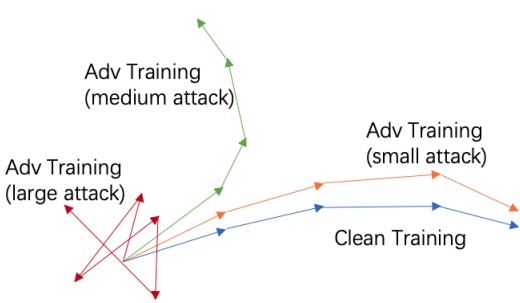

Figure 1: The trajectory of clean training, adversarial training with small/large attack.

Some other literature also supports our conjecture that adversarial training differs from clean training under large attack strength. Gong et al. (2017); Frederickson et al. (2018) show that, as the attack strength increases, the attacked samples are more detectable against clean samples. Such a difference in the training samples leads to the discrepancy between clean and adversarial training. In addition, the trained model from adversarial training is different from the clean model in terms of its Hessian Yao et al. (2018). Furthermore, the neural network capacity required by adversarial training is larger than clean training Xie and Yuille (2019); Bubeck and Sellke (2021).

Note that the "phase transition" phenomenon in this paper refers to an asymptotic behavior rather than a transition at some specific point. In thermodynamics, the phase transition means the state of a substance changes at an exact threshold temperature, e.g., the melting point. In our case, the threshold $\epsilon^*$ refers to an asymptotic order w.r.t. $n$ and model dimension. The behavior of the training trajectory is different under $\epsilon \ll \epsilon^*$ and $\epsilon \gg \epsilon^*$ ($\epsilon/\epsilon^* \to 0$ and $\epsilon/\epsilon^* \to \infty$ in sample size and possibly in data dimension as well).

**Contributions**   The main contributions of this paper can be summarized as follows:

- Directly defining the critical value $\epsilon^*$ requires comparing the whole optimization trajectories between clean training and adversarial training. The whole path comparison can be rather difficult. Thus instead, from theoretical intuitions obtained in simple models, we identify the critical $\epsilon^*$ via a surrogate measure that only relies on the training losses at trajectory destination. In particular, we propose to use clean testing loss in clean training (denote as $R^*(0)$) as a benchmark for adversarial training. If the attack strength $\epsilon$ is small enough so that the (proper) adversarial training loss is smaller than $R^*(0)$, then the adversarial training is similar to the clean training (and vice versa).

- To compute $\epsilon^*$, a naive way is to repeatedly perform the adversarial training from scratch (i.e., initializing from a random start) for each possible $\epsilon$ until the corresponding adversarial training loss hits $R^*(0)$. This could be time-consuming. To speed up the computation, we also propose a method based on adversarial fine-tuning and extrapolation to approximate $\epsilon^*$.

  *Due to the page limit, we postpone many details of approximating $\epsilon^*$ to Appendix A.*

Last but not least, we emphasize that $\epsilon^*$ severs as a diagnosis of the adversarial training, but not a rule of how to choose the attack level. That is, when $\epsilon \ll \epsilon^*$, one can safely assume adversarial training works as well as clean training; when $\epsilon \gg \epsilon^*$, it indicates that the attack strength is too large for the current data set using the current training configuration, and certain remedy is necessary such that the adversarial training performs as good as its clean counterpart. It is beyond our scope to provide a complete solution of how to adjust adversarial training, but some discussions are given in Section 7.

## 2 Other Related Works

**Clean Training in Deep Learning**  Many existing literature studies deep neural networks in clean training. For the training performance, studies such as Du et al. (2019, 2018); Du and Lee (2018) work on the optimization convergence of the empirical training loss. In terms of the testing performance, there are many works showing the good performance of clean training in different aspects, e.g., double-descent phenomenon and benign over-fitting Belkin et al. (2018); Hastie et al. (2019); Ba et al. (2020); Bartlett et al. (2020), implicit regularization Neyshabur et al. (2017b); Neyshabur (2017); Baratin et al. (2020), generalization bounds using PAC-Bayes method or other methods Neyshabur et al. (2017a); Kohler and Langer (2019); Hu et al. (2020); Taheri et al. (2021).

**Adversarial Training**  In the literature, there are several strands of theoretical studies related to adversarial training, e.g., the statistical properties or generalization performance of the global optimizer of adversarial training loss without any consideration of solving the optimization problem Javanmard et al. (2020); Yin et al. (2018); Raghunathan et al. (2019); Min et al. (2020); Zhai et al. (2019); Hendrycks et al. (2019); Chen et al. (2020a); Xing et al. (2021c); Dan et al. (2020), the (local) convergence of adversarial training algorithms under convexity assumptions Sinha et al. (2018); Wang et al. (2019a), the adversarial training loss in deep neural networks (Gao et al., 2019; Zhang et al., 2020b; Wu et al., 2020), and the generalization properties of deep neural networks, e.g. upper bound in Allen-Zhu and Li (2020) and lower bound in Bubeck and Sellke (2021).

## 3 Adversarial Training

To introduce adversarial training, let $l$ denote the loss function and $f_\theta(x)$ be the model with parameter $\theta$. The (population) adversarial loss is defined as $R(\theta, \epsilon) := \mathbb{E}\left[l\left(f_\theta[x + A_\epsilon(f_\theta, x, y)], y\right)\right]$, where $A_\epsilon$ is an attack of strength $\epsilon > 0$ and intends to deteriorate the loss in the following way

$$A_\epsilon(f_\theta, x, y) := \underset{z \in B_p(0, \epsilon)}{\operatorname{argmax}} \{l(f_\theta(x + z), y)\}, \tag{1}$$

where $B_p(x, r)$ is a $\mathcal{L}_p$ ball centering at $x$ with radius $r$. Given $n$ i.i.d. samples $S = \{(x_i, y_i)\}_{i=1}^n$, the adversarial training minimizes the sample version of $R(\theta, \epsilon)$ w.r.t. $\theta$:

$$\widehat{R}(\theta, \epsilon) = \frac{1}{n} \sum_{i=1}^n l\left(f_\theta[x_i + A_\epsilon(f_\theta, x_i, y_i)], y_i\right), \tag{2}$$

and the estimator $\widehat{\theta}(\epsilon)$ aims to minimize $\widehat{R}(\theta, \epsilon)$, and the minimized adversarial training loss is $\widehat{R}_S(\epsilon)$. We rewrite $\widehat{R}(\theta, \epsilon)$ as $\widehat{R}(\theta)$ for simplicity when there is no confusion.

The minimization in (2) is often implemented through an iterative two-step (min-max) update. In the $t$-th iteration, we calculate the adversarial sample $\widetilde{x}_i^{(t)} = x_i + A_\epsilon(f_{\theta^{(t)}}, x_i, y_i)$ based on the current $\theta^{(t)}$, and then update $\theta^{(t+1)}$ based on the gradient of the adversarial training loss while fixing $\widetilde{x}_i^{(t)}$'s with learning rate $\eta_t$. The algorithm runs for $T$ iterations. Note that for some loss function $l$ or model $f_\theta$ (e.g. deep neural networks), there may not be an analytic form for $A_\epsilon$, thus numerical methods, e.g. FGM and PGD, are utilized to approximate $A_\epsilon$. Denote $\theta^{(t)}(\epsilon)$ as the adversarially trained model using attack strength $\epsilon$.

## 4 Intuitions from Simple Models

To obtain insights into the critical strength $\epsilon^*$, we consider the linear regression models. Briefly speaking, we justify that $\epsilon^*$ is in $\Theta(\sqrt{d/n})$ under $\mathcal{L}_2$ attack by comparing the training trajectories of clean and adversarial training, as well as establishing an explicit connection between the critical bound and the generalization error of clean training.

**Simple Linear Regression**  For simple linear regression problem, to measure the difference between the trajectories of adversarial training and clean training, if $\epsilon \ll \sqrt{d/n}$[1], then asymptotically, there is

---

[1]With slightly modification of usual notations, we denote $\epsilon \ll \sqrt{d/n}$ as $\epsilon \log^k n / \sqrt{d/n} \to 0$ for any fixed $k > 0$ to accommodate with some tail probability bounds. The definition of $\epsilon \gg \sqrt{d/n}$ is modified similarly.

no difference between adversarial training and clean training during the training process. Throughout the training (with proper early stopping), we always have that the updating gradient of adversarial training is dominated by the one in clean training. To be specific, we have the following result:

**Theorem 1.** *Assume $Y = \theta_0^\top X + \varepsilon$ where $X \in \mathbb{R}^d$ follows multivariate normal distribution with zero mean and covariance $I_d$, and $\varepsilon$ is a Gaussian noise with constant variance. The true coefficient $\theta_0$ satisfies $\|\theta_0\| = \Theta(1)$ so that $Var(Y) = \Theta(1)$ as well. Consider $\mathcal{L}_2$-adversarial training.*

*When $\epsilon \ll \sqrt{d/n}$, with zero initialization[2], proper learning rate $\eta$ and number of steps $T$, the optimization converges to the global risk minimizer, i.e., $\theta^{(T)}(0) \to \widehat{\theta}(0)$ and $\theta^{(T)}(\epsilon) \to \widehat{\theta}(\epsilon)$. Besides, with probability tending to 1, for all $t \leq T$, the updates in clean and adversarial training have insignificant difference (small attack in Figure 1):*

$$\|\theta^{(t)}(\epsilon) - \theta^{(t)}(0)\|/\|Std(\widehat{\theta}(0))\|_F \to 0.$$

*When $\epsilon \gg \sqrt{d/n}$, with probability tending to 1,*

- *If $\liminf d/n > 0$[3], both $\|\widehat{\theta}(\epsilon)\|$ and $\|\widehat{\theta}(\epsilon)\|/\|\widehat{\theta}(0)\|$ converge to 0 (implying a training trajectory wandering around 0, i.e., large attack in Figure 1). That is, the empirical adversarial risk minimizer is asymptotically zero, giving all training and testing adversarial predictions as zero.*

- *If $\lim d/n = 0$, $\|\widehat{\theta}(\epsilon) - \widehat{\theta}(0)\|/\|Std(\widehat{\theta}(0))\|_F \to \infty$, i.e., the adversarial training trajectory will be statistically different from the clean training process (medium attack in Figure 1).*

- *Such a difference between $\widehat{\theta}(\epsilon)$ and $\widehat{\theta}(0)$ implies the difference in the training trajectories.*

The proof (and details of $\eta$ and $T$) of Theorem 1 is postponed to the appendix.

Theorem 1 implies that $\epsilon^* = \Theta(\sqrt{d/n})$, which gives an appropriate asymptotic order for the critical strength level under linear regression model. However, reproducing such an analysis to find $\epsilon^*$ for complex statistical models is more mathematically involved or even intractable. This motivates us to seek a simpler surrogate measure to identify $\epsilon^*$.

We find that the difference between the clean training loss and clean testing loss can be a proper choice of such a surrogate measure, as illustrated in Theorem 2 below. Using the loss to do comparison helps avoid directly comparing model parameters (i.e., $\theta^{(t)}(\epsilon)$ vs $\theta^{(t)}(0)$), which eases the comparison for complex models such as neural networks in practice. Denote $\Delta$ as the difference between the clean training loss and clean testing loss of clean trained model, i.e., $\Delta = R(\widehat{\theta}(0), 0) - \widehat{R}(\widehat{\theta}(0), 0)$. In practice, one can use $R(\theta^{(T)}(0), 0) - \widehat{R}(\theta^{(T)}(0), 0)$ as an estimate of $\Delta$.

**Theorem 2.** *Under the model setup as in Theorem 1, regardless of the growth of dimension $d$ and sample size $n$, when $\epsilon \ll \sqrt{d/n}$, with high probability,*

$$(\widehat{R}(\widehat{\theta}(0), 0) - \widehat{R}(\widehat{\theta}(\epsilon), \epsilon))/\Delta \to 0.$$

*When $\epsilon \gg \sqrt{d/n}$, the above amount does not go to 0.*

The proof of Theorem 2 is postponed to the appendix.

Both Theorems 1 and 2 describe a phase-transition under linear models with the same phase-transition boundary. This synchronicity justifies our idea of defining critical attack strength $\epsilon^*$ implicitly via the empirical loss and generalization gap $\Delta$. Furthermore, since in real practice mostly we have $R(\widehat{\theta}(0), 0) = \widehat{R}(\widehat{\theta}(0), 0) + \Delta$, $\epsilon^*$ can be defined as

$$\epsilon^* := \sup\{\epsilon \mid \widehat{R}(\widehat{\theta}(\epsilon), \epsilon) \leq R(\widehat{\theta}(0), 0)\}.$$

It is worth mentioning that it is not coincident that both the gap of training losses and the difference of training trajectories occur at the same point. Under linear models, the loss function is always convex for both clean and adversarial training; hence the similarity of gradient descent trajectories highly depends on the similarity of the loss functions.

---

[2]Note that the adversarial loss is not differentiable at $\theta = \mathbf{0}$. But since $\epsilon$ is sufficiently small, it does not affect the convergence.

[3]We exclude the case that $d \in [n - 1, n + 1]$.

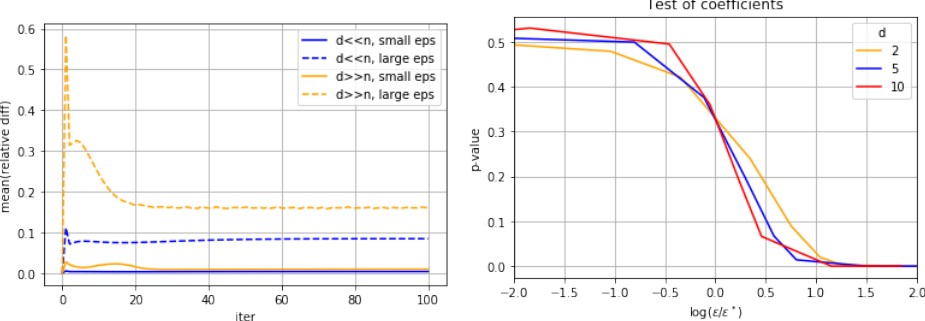

Figure 2: Left: the difference between clean training and adversarial training throughout the training process. The relative difference is calculated as $\|\theta^{(t)}(0) - \theta^{(t)}(\epsilon)\|/\|\theta^{(t)}(0)\|$. The difference gets larger when increasing $\epsilon$. Right: The p-value for hypothesis testing to test whether $\theta^{(T)}(\epsilon)$ follows the same asymptotic distribution as $\theta^{(T)}(0)$. For all setups, the p-value starts to dramatically decrease when $\log(\epsilon/\epsilon^*) \approx 0$.

**Remark 1.** *Theorems 2 and 1 reveal a negative relationship between $\epsilon^*$ and $n$, i.e., larger $n$ leads to a larger discrepancy between adversarial training and cleaning training. However, it does not imply that a larger $n$ hurts the adversarial training. With larger $n$, the adversarial training may act different to clean training, but by the Law of Large Numbers, a large $n$ can force the adversarial training to converge to the correct place.*

**Remark 2.** *The concept of $\epsilon^*$ is for adversarial training, rather than commonly used data augmentation, e.g., via adding Gaussian noise (Reed and MarksII, 1999). The noisy sample is randomly allocated around the original sample. As a result, it does not hurt the training too much when considering the average effect of Gaussian noise, which is not the case for the adversarial attack.*

**Simulation Evidences** We use a simulation study to demonstrate the above observations in the theorems. In particular, we would like to validate numerically (1) the adversarial training parameter is similar to the one in the clean training when $\epsilon \ll \epsilon^*$ and vice versa, and (2) the adversarial training loss is similar to the clean training loss when $\epsilon \ll \epsilon^*$ and vice versa.

To verify (1), we calculate $\|\theta^{(t)}(\epsilon) - \theta^{(t)}(0)\|/\|\theta^{(t)}(0)\|$ for each step $t = 1, ..., 100$, and repeat this experiment 100 times to obtain the average. Due to page limit, we postponed the detailed configurations to the appendix. As shown in Figure 2, when the attack strength is large, there is a great difference between the clean trained model parameters and adversarial trained model parameters. Note that in left panel in Figure 2, we only compare the curves for the same $(d, n)$. The comparison among different $(d, n)$ is not meaningful because the attack strengths are different.

Besides, we conduct hypothesis testing to check whether $\theta^{(T)}(\epsilon)$ is statistically significantly different from $\theta^{(T)}(0)$. We repeat clean training 300 times to obtain the mean and variance of $\theta^{(T)}(0)$, and then calculate the p-value of $\theta^{(T)}(\epsilon)$. The p-value represents "the probability of obtaining test results at least as extreme as the result actually observed, under the assumption that the null hypothesis is correct" (Wikipedia). A close-to-zero p-value means that $\theta^{(T)}(\epsilon)$ is significantly different from $\theta^{(T)}(0)$. We take $n = 100$ and $d \in \{2, 5, 10\}$ in this experiment. The results are shown in the right panel of Figure 2, and one can see that there is a dramatic change in the p-value when $\epsilon$ is around $\epsilon^*$, i.e. $\log(\epsilon/\epsilon^*) \approx 0$.

To verify (2), we also repeat 300 times of clean training to get the distribution of $\widehat{R}(\theta^{(T)}, 0)$ and use this distribution to test whether the adversarial training result is from this distribution. As shown in Figure 3, similar to Figure 2, when the attack strength exceeds $\epsilon^*$, there is a significant difference between the distributions of the losses, and the p-value is close to zero.

**Two-Layer Neural Networks** While the linear regression problem enjoys the above properties, following Ba et al. (2020); Xing et al. (2021b), the two-layer neural network with vanishing initialization can be proved to share similar properties as follows:

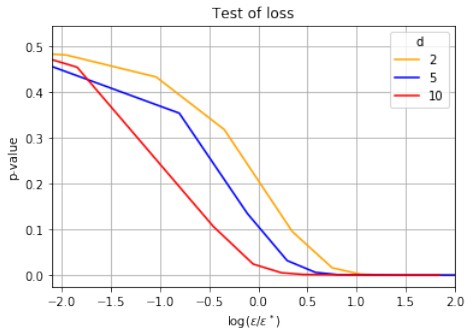

Figure 3: The p-value for hypothesis testing to test whether $\widehat{R}(\theta^{(T)}(\epsilon), \epsilon)$ follows the same asymptotic distribution as $\widehat{R}(\theta^{(T)}(0), 0)$. For all setups, the p-value is close to 0 when $\log(\epsilon/\epsilon^*) \approx 0$.

**Proposition 1** (Two-layer neural networks). *Consider a two-layer neural network with smooth activation function $\phi$*

$$f(x) = \frac{1}{\sqrt{h}} \sum_{j=1}^{h} \phi(x^\top \theta_j) a_j \tag{3}$$

*with $h$ as the number of hidden nodes. When $h \to \infty$, with vanishing initialization of $\theta_j$ and $a_j$, if we fix the second layer ($a_j$) and only train the hidden layer ($\theta_j$), the network parameter satisfies similar (but not the same[4]) property as Theorem 1, and the loss results are the same as Theorem 2.*

The details of Proposition 1 and the main idea of the proof are postponed to the appendix.

## 5 Experiments in Deep Neural Networks

In this section, we study the performance of the proposed $\epsilon^*$ in terms of (1) whether our choice of $\epsilon^*$ is reasonable in deep learning, and (2) what implications the $\epsilon^*$ can bring towards commonly used configurations in deep learning.

### 5.1 General Configurations

Here we describe some general setups in the implementation for both this section and the next section.

For datasets CIFAR-10, CIFAR-100, SVHN, we modified the code of Rice et al. (2020) for our implementation. We keep all the existing configurations (optimizer, neural network architecture, transformer) from the original code. In particular, if there is no specification, we use an SGD optimizer with batch size 128 to train on the full training set for 200 epochs. The learning rate is initialized as 0.1 for CIFAR and 0.01 for SVHN, and is divided by 10 at the 100th and 150th epochs. We consider PreActResNet18 and WideResNet34 as those in Rice et al. (2020). After training 200 epochs, we find the epoch with the smallest adversarial testing loss as the final model for early stopping. For MNIST, we implement a CNN with two convolution layers and two fully connected layers.

Besides, we also use the loss TRADES and MART in Zhang et al. (2019) and Wang et al. (2019b).

### 5.2 Verifying the effectiveness of $\epsilon^*$

In this section, we present some metrics of adversarial training to argue that our choice of $\epsilon^*$ is reasonable. We use CIFAR-10 with PreActResNet18 and $\mathcal{L}_\infty$ in this section.

To obtain $\epsilon^*$, we first run a clean training to get the clean testing loss, then run adversarial training for a wide range of $\epsilon$'s and perform a linear interpolation to obtain $\epsilon^*$. For both clean training and adversarial training, we train from scratch and summarize the results in Table 1. From Table 1, one

---

[4]We cannot directly compare the parameters of the neural networks trained from clean training and adversarial training, but the output predictive models are indeed similar when $\epsilon$ is small enough.

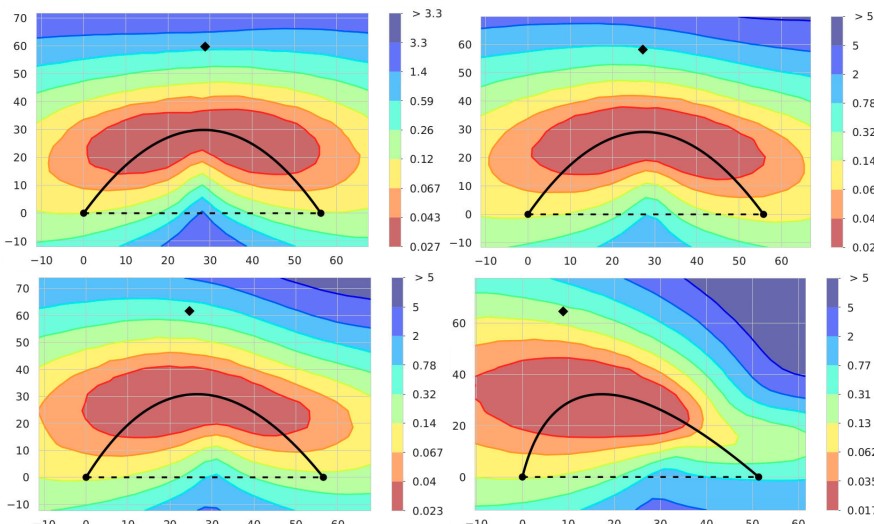

Figure 4: Connectivity between a clean trained model and adversarial robust models in terms of their clean training loss. Upper left: $\epsilon = 0$. Upper right: $\epsilon = 0.5$. Lower left: $\epsilon = 2$. Lower right: $\epsilon = 8$. When increasing $\epsilon$, the right point (the checkpoint for adversarial training) is away from the basin.

can see that the clean testing loss for clean training is $0.1837$. The threshold $\epsilon^*$ is the attack strength such that the adversarial training loss is $0.1837$. From the table, when $\epsilon = 0.5$, the adversarial training loss is $0.1802$, implying that $\epsilon^*$ is approximately $0.5$.

Table 1: Adversarial training in PreActResNet18 using CIFAR-10.

| $\epsilon$ | Epoch | Adv Training | | Adv Testing | |
|---|---|---|---|---|---|
| | | loss | acc | loss | acc |
| 0 | 104 | 0.0761 | 0.9743 | **0.1837** | 0.9405 |
| 0.5/255 | 101 | **0.1802** | 0.9376 | 0.2937 | 0.8977 |
| 1/255 | 102 | 0.2148 | 0.9227 | 0.371 | 0.8723 |
| 2/255 | 102 | 0.3411 | 0.8722 | 0.5218 | 0.8101 |

We conduct comparisons in some aspects, e.g., connectivity, overfitting, and FGM catastrophic overfitting, to show that the adversarial training with $\epsilon \ll \epsilon^*$ is more similar to the clean training compared to stronger attacks.

**Connectivity** The connectivity of deep neural networks aims to answer whether there is a path in the parameter space between two deep neural networks (of the same architecture), such that all the neural networks along this path have good prediction performance. It is a useful tool to study the loss landscape of deep neural networks, e.g., Chao et al. (2020). Good connectivity implies that the two neural networks are in the same "basin" of the loss.

To numerically figure out a path, we use the method in Garipov et al. (2018). In order to have a graphical presentation, we take num_bend as 3, i.e., besides the start and the end neural networks, there is only one neural network to be trained in the path parameters. Figure 4 shows the connectivity between the clean trained model and the robust model. In each subfigure, the left point represents the clean model at the 101st epoch, the right point is the adversarially trained model at the 101st epoch, and the upper point is the extra checkpoint to be trained. We take the 101st epoch because the adversarial training generally achieves the best adversarial testing loss around the 101st epoch. These three points together determine the black curve in the figure, and they are in the same 2D plane. We calculate the clean training loss for other points in this plane to obtain the contour.

From Figure 4, one can observe that when $\epsilon = 0$ and $\epsilon = 0.5$, the low-loss region is almost symmetric to the lower two points. The middle part is also well connected, i.e., most of the connecting path belongs to the low-loss region (red area). When $\epsilon > 0.5$, the low-loss region gradually shifts towards the clean model. This indicates that the adversarial training gradually moves out of the "basin".

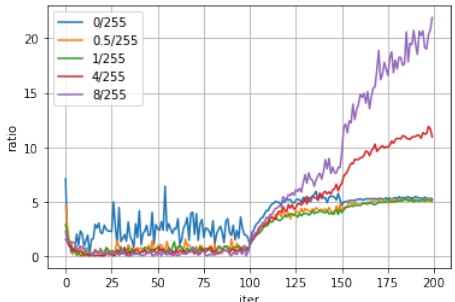

Figure 5: Overfitting in clean training and adversarial training. We use the ratio of generalization error over the one at the 101st epoch to examine the overfitting level, where generalization error is the difference between adversarial training and testing loss. When $\epsilon \leq 1$, this ratio is small. Results from other attack strengths, e.g. 2/255 and 16/255, are not included in this figure to keep the plot clear, and they do not alter the main observations.

**Generalization and Overfitting**   In the literature, it is observed that in deep neural networks with clean training, overfitting training data does not affect the generalization performance. Some studies, e.g., Belkin et al. (2018, 2019a); Li et al. (2021); Chatterji et al. (2021), provide theoretical justifications towards this phenomenon.

Our numerical results show no harmful overfitting in clean training, and the overfitting in adversarial training with $\epsilon^*$ is also not severe. As shown in Figure 5, when $\epsilon$ gets larger, the ratio (generalization gap)/(generalization gap at the 101st epoch) increases after the 100th epoch.

**FGSM and Catastrophic Overfitting**   Based on Andriushchenko and Flammarion (2020), FGSM (i.e., the FGM approximation of $\mathcal{L}_\infty$-PGD attack) leads to catastrophic overfitting when attack strength is large, yielding an almost-zero adversarial testing accuracy using $\mathcal{L}_\infty$-PGD attack. Since FGM is only an approximation of PGD, the FGM attack is not as strong as the PGD attack, and its adversarial testing performance under PGD attack is expected to be worse. However, for $\mathcal{L}_\infty$ attack, the performance drop in testing accuracy is far more severe, and Andriushchenko and Flammarion (2020) reveals some threats in FGSM which cause this phenomenon. This catastrophic overfitting is a second type of overfitting different from the one discussed in the previous paragraph.

Based on our intuition of $\epsilon^*$, the above concern does not hurt the training process when $\epsilon < \epsilon^*$. First, since $\epsilon$ is small, the difference between FGM and PGD is small. Second, since adversarial training is similar to clean training and clean training does not suffer from severe over-fitting problems, those problems can be avoided. To verify this, we train CIFAR-10 using 10000/50000 samples with PGD/FGM adversarial training in different levels of $\mathcal{L}_\infty$ attacks. We take $\epsilon$ as 1/255, 2/255, 4/255, 8/255, 12/255, 16/255 to see whether it obtains a large adversarial testing loss with an almost-zero adversarial testing accuracy.

The results are summarized in Table 2. The column "$\epsilon^*$" shows the critical $\epsilon^*$, and the column "Unstable $\epsilon$" is the minimal attack strength when FGSM starts the catastrophic over-fitting. One can see that $\epsilon^*$ is always smaller than the unstable threshold. This is intuitive because adversarial training is still similar to clean training with $\epsilon \approx \epsilon^*$. On the other hand, this also implies that one can use FSGM to search for $\epsilon^*$.

Table 2: Catastrophic overfitting in FGSM adversarial training in CIFAR-10 using PreActResNet18.

| n | norm | $\epsilon^*$ | Unstable $\epsilon$ |
|---|---|---|---|
| 10K | $\mathcal{L}_\infty$ | 3.3/255 | 12/255 |
| 50K | $\mathcal{L}_\infty$ | 0.5/255 | 8/255 |

## 5.3 Observations in common settings

The previous section conducts numerical experiments to justify our choice of $\epsilon^*$. In this section, we use this method in various datasets and neural network architectures to provide more insights and study how $\epsilon^*$ is affected by these factors.

Table 3: The value of $\epsilon^*$ in different datasets, neural network architectures, and training sample size. "PAResNet" and "WResNet" refer to "PreActResNet" and "WideResNet" respectively to save the margin.

| Dataset | Architecture | $n$ | $\epsilon^*(\mathcal{L}_\infty)$ | Dataset | Architecture | $n$ | $\epsilon^*(\mathcal{L}_\infty)$ |
|---|---|---|---|---|---|---|---|
| CIFAR-10 | PAResNet18 | 50K | 0.5/255 | MNIST | MLP(128) | 50K | 48.0/255 |
| CIFAR-10 | PAResNet18 | 10K | 3.3/255 | MNIST | MLP(16) | 50K | 26.1/255 |
| CIFAR-10 | WResNet34-1 | 50K | 1.1/255 | SVHN | PAResNet18 | 50K | 4.9/255 |
| CIFAR-10 | WResNet34-10 | 50K | 1.1/255 | | | | |
| CIFAR-100 | PAResNet18 | 25K | 4.0/255 | | | | |
| CIFAR-100 | PAResNet18 | 50K | 1.0/255 | | | | |

### 5.3.1 General Observations

An important observation in Table 3 is that the commonly used attack strengths are greater than $\epsilon^*$. In the literature, we usually use $\mathcal{L}_\infty$ attack with strength $8/255$ for CIFAR-10/100 and SVHN, and $0.3$ for MNIST. From our experiments, we observe that using all the 50000 training samples, the corresponding $\epsilon^*$ for CIFAR-10/100, SVHN, and MNIST are $0.5/255$, $1.0/255$, $4.9/255$, $48.0/255$.

### 5.3.2 More Detailed Observations

**Wide neural networks** Different neural network architectures lead to different $\epsilon^*$. For MNIST, using a simple neural network of two convolution layers and two fully connected layers, when there are 16 hidden nodes for the first FC layer, $\epsilon^*$ is 26.1/255. When there are 128 hidden nodes, it becomes 48/255. This observation verifies the argument in the existing literature that wider neural network structures are essential in adversarial training to enlarge the model capacity Xie and Yuille (2019); Author (2021). For CIFAR-10, $\epsilon^*$ changes little when expanding the width of the neural network from 1 to 10, implying that WideResNet34-10 may not be sufficient for adversarial training. Note that one need to be cautious when enlarging the network size, as wider neural networks tend to overfit the data even in clean training.

**Improved adversarial loss** Some literature tries to improve the adversarial training process via improving the loss. We would like to examine how $\epsilon^*$ changes along these loss functions as well. We run TRADES and MART in CIFAR-10 with PreActResNet18 for this experiment, and the $\epsilon^*$ are 0.7/255 and 0.8/255, respectively. Although 0.7/255 and 0.8/255 are larger than 0.5/255 (i.e., $\epsilon^*$ for vanilla adversarial training; See Table 3), they are still less than 8/255.

**Training size** From Table 3, for both CIFAR-10 and CIFAR-100, a larger training size indicates a smaller $\epsilon^*$. There are two reasons for this phenomenon. First, when enlarging the sample size, the generalization gap gets smaller; thus, $\epsilon^*$ is smaller. Second, as observed in Author (2021), when increasing the training size, the neural network has a larger norm, implying that a larger model capacity is needed to fit the attacked samples. As a result, the neural network architecture cannot handle the adversarial training properly, so the corresponding $\epsilon^*$ is smaller.

## 6 Faster Approximation of $\epsilon^*$

To compute $\epsilon^*$, a naive way is to repeatedly perform the adversarial training for each possible $\epsilon$ until it hits the threshold. This could be very time-consuming when solving the adversarial training from scratch (i.e., initializing from a near-zero random start). For instance, it takes 20 minutes to train the CIFAR-10 dataset for clean training but takes 10 hours to complete adversarial training for a given $\epsilon$. Hence we propose to speed up this process via a linear extrapolation approximation.

*Due to the page limit, the intuition, potential difficulties, and final algorithm for approximating $\epsilon^*$ are postponed to Section A in the appendix.* Briefly speaking, due to good connectivity between the clean and adversarial training when $\epsilon \ll \epsilon^*$ as in Figure 4, we consider using adversarial fine-tuning (Chen et al., 2020b) to replace the whole adversarial training (from scratch) process in the grid search for $\epsilon^*$. To overcome the potential over-fitting and learning rate tuning problems, we conduct adversarial fine-tuning under similar $\epsilon$'s and use extrapolation to approximate $\epsilon^*$. Since we are using extrapolation rather than interpolation, we also provide theoretical support to justify the correctness

Table 4: Performance of fine tuning. $\widehat{\epsilon}$(FGM,$a$) represents that the adversarial fine-tuning uses FGM attack with $a$ runs of fine tuning.

| n | norm | $\epsilon^*$ | $\widehat{\epsilon}$(PGD,1) | $\widehat{\epsilon}$(PGD,3) | $\widehat{\epsilon}$(FGM,1) | $\widehat{\epsilon}$(FGM,3) |
|---|---|---|---|---|---|---|
| 10000 | $\mathcal{L}_\infty$ | 3.3/255 | 4.7/255 | 2.5/255 | 1.7/255 | 2.1/255 |
| 50000 | $\mathcal{L}_2$ | 24.8/255 | 20.1/255 | 17.9/255 | 18.4/255 | 17.4/255 |
| 50000 | $\mathcal{L}_\infty$ | 0.5/255 | 0.58/255 | 0.52/255 | 0.53/255 | 0.59/255 |

of the algorithm. The numerical results are shown in Table 4. One can see that the proposed algorithm gives a good estimate of $\epsilon^*$.

# 7 What Can We Do If $\epsilon \gg \epsilon^*$?

Our paper focuses on how to determine $\epsilon^*$ and the consequence of $\epsilon \gg \epsilon^*$, and it is beyond our scope to study how to adjust adversarial training when $\epsilon \gg \epsilon^*$. We provide some potential solutions.

In general, when $\epsilon \gg \epsilon^*$, one needs to overcome a series of potential problems in adversarial training. In the literature, methods such as MART Wang et al. (2019b), Dynamic Wang et al. (2019a), FAT Zhang et al. (2020a), HAT Rade and Moosavi-Dezfooli (2021), smoothing Xie et al. (2020); Xing et al. (2021b) can overcome some of the problems. However, they may not be sufficient to resolve every problem caused by the fundamental gap between adversarial training and cleaning training revealed by this work. Alternatively, we suggest two other ways: (1) adjusting the neural network architecture to enlarge $\epsilon^*$, and (2) utilizing more information/data to force adversarial training to converge to the correct place.

For (1), in our numerical experiment, enlarging the size of the neural network can make $\epsilon^*$ larger. If controlling the over-fitting issue properly, this can be a solution to improve adversarial robustness. Similarly, empirical experiments in various literature (e.g., Xie and Yuille, 2019; Rice et al., 2020; Gowal et al., 2021) show that wider neural networks lead to better performance. Some other studies (e.g., Huang et al., 2021) also study how the neural network architecture affects robustness.

For (2), there are several ways to utilize more information from the data. For example, Gowal et al. (2021) trains a clean classifier and an unlabeled data generator to generate extra synthetic data, which facilitate the adversarial training. This framework does not introduce any new data source but improves adversarial robustness, implying that vanilla adversarial training overlooks some information from the data. Although $\epsilon^*$ may be smaller than the actual $\epsilon$, using more information/data can force the training to converge to the correct place. Similar studies can be found in Carmon et al. (2019); Xing et al. (2021c).

# 8 Conclusion

Observing the great gap between empirical results in adversarially robust models and the theoretical studies in this area, we conjecture that adversarial training acts differently from our common understanding of clean training in deep learning. Through intuitions in simple statistical models, we design a metric of similarity to determine whether adversarial training is "similar" to clean training or not. Our results reveal that the commonly used adversarial training setups in literature take a large attack strength so that it is different from clean training given the current neural network architectures and data. We reveal some potential factors which affect $\epsilon^*$. Besides, since adversarial training from scratch is time-consuming, we propose to use adversarial fine-tuning and extrapolation to do approximate $\epsilon^*$. Such a method can reduce 80% running time compared to adversarial training from scratch while leading to reasonable estimates.

A future direction of this work is to theoretically understand how the neural network architecture affects $\epsilon^*$, and based on which, to provide proper guidance on architecture selection that accommodates stronger attacks (e.g., the commonly used 8/255 attack strength in practice).

## Acknowledgements

This project is partially supported by NSF-SCALE MoDL (2134209) and ONR N00014-22-1-2680.

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
