## A    Speed up the Approximation

In this section, we state the possibility (feasibility) to speed up the approximation using adversarial fine-tuning, and describe what potential problems we have in this method, as well as our solution and the numerical results.

**Feasibility of Adversarial Fine-Tuning**    One may question whether fine-tuning (i.e., initializing from an estimated model trained under a similar setup) can achieve our goal since there is no guarantee that adversarial training from scratch converges to the same places as adversarial fine-tuning. We use connectivity to illustrate the feasibility of the fine-tuning method. Based on Figure 4 in Section 5.2, when taking a small $\epsilon$, the connectivity between the clean model and adversarially robust model (trained from scratch) is good. This good connectivity implies that solving $\epsilon^*$ via adversarial training from scratch and adversarial fine-tuning lead to similar estimates.

**A Practical Problem**    Following Chen et al. (2020b), one can obtain an adversarially robust model via fine-tuning. However, some simple trials reveal two potential problems:

1. if we fine-tune from a clean model, i.e., $\theta^{(0)}(\epsilon) = \theta^{(t)}(0)$, the learning rate schedule that works for small $\epsilon$ is not suitable for large $\epsilon$;

2. if we sequentially fine-tune the model, i.e., $\theta^{(0)}(\epsilon_{i+1}) = \theta^{(t)}(\epsilon_i)$ for an increasing sequence of $\epsilon_i$'s, it can be easily trapped by a bad local minimum which overfits the training data and yields zero adversarial training loss for large $\epsilon$.

Consequently, it is hard to provide a fully automatic algorithm to solve the exact $\epsilon^*$ without manually tuning the learning rate schedule.

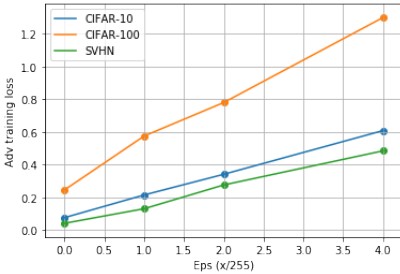

Figure A.1: The adversarial training loss (with early stopping) for three real datasets. The loss is approximately a linear function in the attack strength.

**Solution**    To overcome the above practical problem, we propose to use adversarial fine-tuning for only a small number of $\epsilon$'s whose values are close, such that the same learning rate schedule can be adopted. After obtaining the adversarial fine-tuning results for these $\epsilon$'s, we then use regression extrapolation to approximate $\epsilon^*$. This process can significantly reduce the computation and time cost of finding $\epsilon^*$.

The success of the above approximation idea relies on how accurate the extrapolation is. By the definitions of adversarial attack and adversarially robust models, the adversarial risk is a monotone

increasing function w.r.t. the attack strength $\epsilon$. If we plot $\epsilon$ against the corresponding adversarial training loss $\widehat{R}(\widehat{\theta}(\epsilon), \epsilon)$ (trained from scratch), the curve is almost a linear function as in Figure A.1. Based on this observation, linear (or isotonic if necessary) extrapolation with adversarial fine-tuning gives a reasonable approximation.

The following Algorithm A.1 describes the above idea:

---

**Algorithm A.1** Solve $\epsilon^*$

---

Input: the trained clean model $\widehat{\theta}(0)$, data $\{(x_i, y_i)\}_{i=1}^n$, learning rate schedule $\{\eta_t\}$, attack strength $\{\epsilon_j\}$.
**for** $\epsilon = \epsilon_1, \epsilon_2, \epsilon_3, ...$ **do**
    Use adversarial fine tuning with $\epsilon$ to obtain the minimal adversarial training loss when minimizing adversarial testing loss. Denote the adversarial model as $\widehat{\theta}(\epsilon)$.
**end for**
Use $(\epsilon, \widehat{R}(\widehat{\theta}(\epsilon), \epsilon))$ for $\epsilon = 0, \epsilon_1, \epsilon_2, \epsilon_3, ...$ to find an approximation $\widehat{\epsilon}$ of $\epsilon^*$.

---

Algorithm A.1 illustrates the general idea, while we do not explicitly specify the exact number of fine-tunes and the type of approximation method used. One can use the clean model and one fine-tuned single $\epsilon_1$ to conduct the linear extrapolation, or several models for linear regression or nonlinear extrapolation (e.g., monotone splines). For the experiment presented in this paper, we choose linear regression.

The following proposition illustrates the effectiveness of Algorithm A.1:

**Proposition 2.** *Consider the linear models in Theorem 2.*

*Adversarial fine tuning satisfies that, if $\|\theta^{(t)}\| \leq r$ for all $t$ for some $r > 0$, then*

$$
\mathbb{E}\left[\min_{t=1,...,T} \widehat{R}(\theta^{(t)}) - \widehat{R}(\widehat{\theta}) \Big| S\right] \leq \frac{\mathbb{E}\|\theta^{(0)} - \widehat{\theta}\|^2}{2\eta T} - \frac{\mathbb{E}[\|\theta^{(T)} - \widehat{\theta}\|^2 | S]}{2\eta T} + O\left(\frac{\eta(L_r)^2}{2}\right),
$$

*where $L_r$ is the Lipschitz constant of $\widehat{R}$ for $\theta \in B_2(0, r)$.*

*Given sufficiently small $\eta$ and large $T$, the above optimization error will be negligible, then the approximation error involved in $\widehat{\epsilon}$ is $o(\epsilon^*)$ if all $\epsilon$ and $\epsilon^*$ are $o(1)$.*

Proposition 2 shows the convergence of the proposed algorithm, and the proof is in Section C.

Proposition 2 itself is applicable to any proper initialization of fine-tuning. Compared to training from scratch, since the trained clean model already has a small adversarial training loss, taking $\widehat{\theta}(0)$ as the initialization of Algorithm A.1, the number of iterations used in the fine-tuning is much less.

Besides, Algorithm A.1 only requires small-$\epsilon$ adversarial training, based on our observations in Section 5 for the catastrophic overfitting, using FGM is also a fast and effective approximation if we replace it with PGD attack in the adversarial fine-tuning when $\epsilon = O(\epsilon^*)$.

**Numerical Illustration**    This section aims to verify that (1) the proposed method with extrapolation is still reasonable, and (2) using FGM leads to a good estimate of $\epsilon^*$.

After obtaining the clean model, we use 40 epochs to do the adversarial fine-tuning for each $\epsilon$, so there will be at least 80% reduction on the computation time compared to adversarial training from scratch. We use the same base learning rate as the one in clean training. We use an increasing and decreasing learning rate schedule in the fine-tuning to adjust the learning rate. The detailed learning rate schedule and the choices of $\epsilon$ we take (together with the loss values) are in Section B.

The results are summarized in Table 4. We consider both PGD and FGM in our experiments. In Table 4, we present the two adversarial fine-tuning algorithms using PGD and FGM, respectively, and the all one- and three-points approximation for $\epsilon^*$. All the settings lead to good approximation.

# B  Experiment Details

## B.1  Simulation in Section 4

Table B.1 shows the the detailed configurations for the experiment for the left panel of Figure 2.

Table B.1: Detailed parameter setups for Figure 2

| Regime | $d$ | $n$ | $\epsilon^*$ | Small $\epsilon$ | Large $\epsilon$ |
|--------|-----|-----|--------------|------------------|-------------------|
| $d \gg n$ | 20 | 1000 | 0.02 | 0.01 | 0.2 |
| $d \ll n$ | 2000 | 10 | 15 | 1 | 20 |

## B.2  Details for $\epsilon^*$

To obtain $\epsilon^*$ in Table 2, 4, and 3, we use adversarial training to train on different $\epsilon$s from scratch, then use linear interpolation to approximate $\epsilon^*$. We paste the detailed training results here in Table B.2, B.3, B.4, B.5, and B.6. Table B.2 collects the results for CIFAR-10 dataset using different training sizes, batch sizes, and attack norms. Table B.3 is the result for other datasets. Table B.4 is the result for CIFAR-10 using wide neural networks, and Table B.5 is the result of CIFAR-10 using Adam optimizer, or TRADES/MART loss. Table B.6 summarizes the performance of FGM in adversarial training using CIFAR-10.

Table B.2: Adversarial training in CIFAR-10 (PreActResNet) using different training sample size (10000, 50000), batch size (128, 256), attack norm ($\mathcal{L}_\infty$, $\mathcal{L}_2$).

| $N$ | $\epsilon$(/255) | Batch size | Norm | Train Adv loss | Train Adv acc | Test Adv loss | Test Adv acc |
|-----|------------------|------------|------|----------------|---------------|---------------|--------------|
| 10000 | 0 | 128 | $\mathcal{L}_\infty$ | 0.0403 | 0.9894 | 0.441 | 0.8717 |
| 10000 | 1 | 128 | $\mathcal{L}_\infty$ | 0.1586 | 0.9462 | 0.879 | 0.734 |
| 10000 | 2 | 128 | $\mathcal{L}_\infty$ | 0.2199 | 0.9269 | 1.1723 | 0.6555 |
| 10000 | 8 | 128 | $\mathcal{L}_\infty$ | 1.2186 | 0.5114 | 1.7621 | 0.3723 |
| 10000 | 32 | 128 | $\mathcal{L}_2$ | 0.0777 | 0.9778 | 0.9131 | 0.7396 |
| 10000 | 64 | 128 | $\mathcal{L}_2$ | 0.2001 | 0.9339 | 1.1616 | 0.6621 |
| 10000 | 96 | 128 | $\mathcal{L}_2$ | 0.2681 | 0.908 | 1.4464 | 0.57 |
| 10000 | 128 | 128 | $\mathcal{L}_2$ | 0.3494 | 0.8743 | 1.5258 | 0.5263 |
| 10000 | 256 | 128 | $\mathcal{L}_2$ | 0.8098 | 0.6826 | 1.8491 | 0.3773 |
| 10000 | 512 | 128 | $\mathcal{L}_2$ | 1.6061 | 0.3789 | 2.0156 | 0.2782 |
| 50000 | 0 | 128 | $\mathcal{L}_\infty$ | 0.0761 | 0.9743 | 0.1837 | 0.9405 |
| 50000 | 0.5 | 128 | $\mathcal{L}_\infty$ | 0.1802 | 0.9376 | 0.2937 | 0.8977 |
| 50000 | 1 | 128 | $\mathcal{L}_\infty$ | 0.2148 | 0.9227 | 0.371 | 0.8723 |
| 50000 | 2 | 128 | $\mathcal{L}_\infty$ | 0.3411 | 0.8722 | 0.5218 | 0.8101 |
| 50000 | 4 | 128 | $\mathcal{L}_\infty$ | 0.6082 | 0.7631 | 0.8079 | 0.6972 |
| 50000 | 8 | 128 | $\mathcal{L}_\infty$ | 1.1102 | 0.5645 | 1.238 | 0.5291 |
| 50000 | 0 | 256 | $\mathcal{L}_\infty$ | 0.0263 | 0.9924 | 0.1823 | 0.9469 |
| 50000 | 1 | 256 | $\mathcal{L}_\infty$ | 0.1769 | 0.9389 | 0.4179 | 0.8582 |
| 50000 | 2 | 256 | $\mathcal{L}_\infty$ | 0.2779 | 0.8972 | 0.5858 | 0.8022 |
| 50000 | 4 | 256 | $\mathcal{L}_\infty$ | 0.5181 | 0.7991 | 0.8656 | 0.6924 |
| 50000 | 32 | 128 | $\mathcal{L}_2$ | 0.2149 | 0.9212 | 0.3739 | 0.8675 |
| 50000 | 64 | 128 | $\mathcal{L}_2$ | 0.34 | 0.8702 | 0.5362 | 0.8038 |
| 50000 | 96 | 128 | $\mathcal{L}_2$ | 0.4698 | 0.8159 | 0.6775 | 0.7484 |
| 50000 | 128 | 128 | $\mathcal{L}_2$ | 0.5963 | 0.7644 | 0.8038 | 0.7012 |

## B.3  Adversarial Fine-tuning

Similar to solving $\epsilon^*$, when using adversarial fine-tuning, we also runs the fine-tuning for some different $\epsilon$s. The detailed results for each $\epsilon$ are summarized in Table B.7.

Table B.3: Adversarial training using other datasets (CIFAR-100, SVHN) (PreActResNet18).

| Dataset | $N$ | $\epsilon$(/255) | Adv train loss | Adv train acc | Adv test loss | Adv test acc |
|---|---|---|---|---|---|---|
| CIFAR-100 | 25000 | 0 | 0.2459 | 0.935 | 1.2872 | 0.667 |
| CIFAR-100 | 25000 | 1 | 0.5751 | 0.8322 | 2.1848 | 0.4938 |
| CIFAR-100 | 25000 | 2 | 0.7798 | 0.7737 | 2.5653 | 0.4029 |
| CIFAR-100 | 25000 | 4 | 1.2994 | 0.634 | 3.0408 | 0.312 |
| CIFAR-100 | 25000 | 8 | 2.7874 | 0.2804 | 3.5278 | 0.1899 |
| CIFAR-100 | 50000 | 0 | 0.4647 | 0.8617 | 0.9047 | 0.7448 |
| CIFAR-100 | 50000 | 1 | 0.8917 | 0.7335 | 1.4699 | 0.5941 |
| CIFAR-100 | 50000 | 2 | 1.383 | 0.6041 | 1.7852 | 0.5126 |
| CIFAR-100 | 50000 | 4 | 1.7344 | 0.517 | 2.2901 | 0.4032 |
| CIFAR-100 | 50000 | 8 | 2.6004 | 0.3356 | 2.9453 | 0.2834 |
| SVHN | 50000 | 0 | 0.0006 | 1 | 0.2303 | 0.9438 |
| SVHN | 50000 | 2 | 0.0016 | 0.9999 | 0.7909 | 0.8177 |
| SVHN | 50000 | 4 | 0.0931 | 0.9648 | 1.4166 | 0.6563 |
| SVHN | 50000 | 8 | 0.7251 | 0.7419 | 1.6793 | 0.4806 |

Table B.4: The performance of adversarial training in CIFAR-10 (WideResNet34, $N = 50000$) using different width factor.

| Width | $\epsilon$(/255) | Adv train loss | Adv train acc | Adv test loss | Adv test acc |
|---|---|---|---|---|---|
| 1 | 0 | 0.0476 | 0.986 | 0.2261 | 0.934 |
| 1 | 1 | 0.2061 | 0.9266 | 0.4645 | 0.8402 |
| 1 | 2 | 0.3782 | 0.8594 | 0.6365 | 0.7723 |
| 10 | 0 | 0.0416 | 0.9872 | 0.1505 | 0.954 |
| 10 | 1 | 0.1307 | 0.9546 | 0.3311 | 0.8892 |
| 10 | 2 | 0.2759 | 0.8997 | 0.4603 | 0.8329 |

Table B.5: The performance of adversarial training in CIFAR-10 (PreActResNet18, $N = 50000$) using other optimizer (Adam) or loss function (TRADES, MART).

| Name | $\epsilon$(/255) | Adv train loss | Adv train acc | Adv test loss | Adv test acc |
|---|---|---|---|---|---|
| Adam | 0 | 0.0307 | 0.9896 | 0.3554 | 0.9204 |
| Adam | 1 | 0.0767 | 0.972 | 0.6983 | 0.8224 |
| Adam | 2 | 0.2832 | 0.8883 | 0.9108 | 0.7257 |
| Adam | 4 | 0.9711 | 0.6289 | 1.1258 | 0.5972 |
| MART | 0 | 0.432 | 0.9195 | 0.3411 | 0.8856 |
| MART | 1 | 0.5989 | 0.8381 | 0.5627 | 0.8032 |
| MART | 2 | 0.8447 | 0.7687 | 0.7675 | 0.727 |
| TRADES | 0 | 0.1681 | 0.9613 | 0.2857 | 0.9044 |
| TRADES | 1 | 0.3212 | 0.825 | 0.5536 | 0.8111 |
| TRADES | 2 | 0.3007 | 0.7767 | 0.8246 | 0.7094 |

## B.4 Fine-tuning learning rate

Figure B.1 shows the learning rate schedule used for adversarial fine-tuning. The values in Figure B.1 are multiplied by `lr_max` in training.

Table B.6: FGM adversarial training for CIFAR-10.

| $N$ | $\epsilon$ (/255) | Norm | Adv train loss | Adv train acc | Adv test loss | Adv test acc |
|---|---|---|---|---|---|---|
| 10000 | 1 | $\mathcal{L}_\infty$ | 0.1418 | 0.958 | 0.8212 | 0.7483 |
| 10000 | 2 | $\mathcal{L}_\infty$ | 0.2066 | 0.9297 | 1.1251 | 0.6587 |
| 10000 | 4 | $\mathcal{L}_\infty$ | 0.5863 | 0.7665 | 1.6388 | 0.4856 |
| 10000 | 8 | $\mathcal{L}_\infty$ | 0.9963 | 0.5905 | 2.0131 | 0.3419 |
| 10000 | 12 | $\mathcal{L}_\infty$ | 0.265 | 0.9151 | 31.7608 | 0 |
| 10000 | 16 | $\mathcal{L}_\infty$ | 0.3133 | 0.8996 | 36.0938 | 0 |
| 10000 | 20 | $\mathcal{L}_\infty$ | 0.1067 | 0.9666 | 36.4307 | 0 |
| 10000 | 32 | $\mathcal{L}_\infty$ | 0.1253 | 0.9585 | 33.5073 | 0 |
| 50000 | 1 | $\mathcal{L}_\infty$ | 0.1785 | 0.9381 | 0.3863 | 0.8682 |
| 50000 | 2 | $\mathcal{L}_\infty$ | 0.2727 | 0.9005 | 0.5578 | 0.8022 |
| 50000 | 4 | $\mathcal{L}_\infty$ | 0.4683 | 0.8195 | 0.8726 | 0.6839 |
| 50000 | 8 | $\mathcal{L}_\infty$ | 0.3184 | 0.8981 | 29.4849 | 0 |
| 50000 | 12 | $\mathcal{L}_\infty$ | 0.2812 | 0.9098 | 29.5383 | 0 |
| 50000 | 16 | $\mathcal{L}_\infty$ | 0.2475 | 0.9205 | 39.2974 | 0 |
| 50000 | 20 | $\mathcal{L}_\infty$ | 0.2216 | 0.9293 | 39.3613 | 0 |
| 50000 | 32 | $\mathcal{L}_\infty$ | 0.2144 | 0.9315 | 33.7759 | 0 |

Table B.7: The adversarial training loss using adversarial fine-tuning.

| Attack | $N$ | $\epsilon$(/255) | Norm | Adv loss | Attack | $N$ | $\epsilon$(/255) | Norm | Adv loss |
|---|---|---|---|---|---|---|---|---|---|
| None | 10000 | 0 | - | 0.0403 | | | | | |
| PGD | 10000 | 0.25 | $\mathcal{L}_\infty$ | 0.0426 | FGM | 10000 | 0.25 | $\mathcal{L}_\infty$ | 0.0893 |
| PGD | 10000 | 0.5 | $\mathcal{L}_\infty$ | 0.0829 | FGM | 10000 | 0.5 | $\mathcal{L}_\infty$ | 0.1562 |
| PGD | 10000 | 1 | $\mathcal{L}_\infty$ | 0.2008 | FGM | 10000 | 1 | $\mathcal{L}_\infty$ | 0.2289 |
| None | 50000 | 0 | - | 0.0761 | | | | | |
| PGD | 50000 | 0.25 | $\mathcal{L}_\infty$ | 0.1163 | FGM | 50000 | 0.25 | $\mathcal{L}_\infty$ | 0.1183 |
| PGD | 50000 | 0.5 | $\mathcal{L}_\infty$ | 0.1642 | FGM | 50000 | 0.5 | $\mathcal{L}_\infty$ | 0.1739 |
| PGD | 50000 | 1 | $\mathcal{L}_\infty$ | 0.288 | FGM | 50000 | 1 | $\mathcal{L}_\infty$ | 0.2425 |
| PGD | 50000 | 5 | $\mathcal{L}_2$ | 0.0999 | FGM | 50000 | 5 | $\mathcal{L}_2$ | 0.0958 |
| PGD | 50000 | 10 | $\mathcal{L}_2$ | 0.1273 | FGM | 50000 | 10 | $\mathcal{L}_2$ | 0.1321 |
| PGD | 50000 | 15 | $\mathcal{L}_2$ | 0.1655 | FGM | 50000 | 15 | $\mathcal{L}_2$ | 0.1673 |

Figure B.1: Learning rate multiplier used in adversarial fine-tuning.

# C   Proofs

## C.1   Theorem 1

To prove Theorem 1, we first figure out $\|Std(\widehat{\theta}(0))\|_F$ in the following lemma, then present the main proof.

**Lemma 1.** *Under the data generation model in Theorem 1, when $n - d - 1 > 0$,*

$$Var(\widehat{\theta}(0)) = \frac{\sigma^2 I_d}{n - d - 1},$$

*and when $d - n - 1 > 0$,*

$$tr(Var(\widehat{\theta}(0))) = \frac{\sigma^2 n}{d - n - 1}.$$

*Consequently,*

$$\|Std(\widehat{\theta}(0))\|_F = \begin{cases} \Theta(\sqrt{d/n}) & d \ll n \\ \Theta(1) & d \asymp n \\ \Theta(\sqrt{n/d}) & d \gg n \end{cases}.$$

*Proof of Lemma 1.* From the definition of $Std$ and $Var$, we know that

$$\|Std(\widehat{\theta}(0))\|_F = \sqrt{tr(Std(\widehat{\theta}(0))Std(\widehat{\theta}(0)))} = \sqrt{tr(Var(\widehat{\theta}(0)))}.$$

Since we use linear regression with Gaussian design, the closed-form solution of $\widehat{\theta}(0)$ exists and its variance can be directly computed.

Denote $X_n$ as the data matrix and $Y_n$ as the corresponding response vector. Also define $\varepsilon_n$ as the noise vector.

***Case 1,*** $n - d - 1 > 0$*:* we know that

$$\widehat{\theta}(0) = (X_n^\top X_n)^{-1} X_n^\top Y_n = \theta_0 + (X_n^\top X_n)^{-1} X_n^\top \varepsilon_n.$$

As a result, the variance of $\widehat{\theta}(0)$ becomes

$$
\begin{aligned}
&Var(\widehat{\theta}(0)) \\
=\ & Var((X_n^\top X_n)^{-1} X_n \varepsilon_n) \\
=\ & \mathbb{E}(X_n^\top X_n)^{-1} X_n^\top \varepsilon_n \varepsilon_n^\top X_n (X_n^\top X_n)^{-1} - \mathbb{E}\left[(X_n^\top X_n)^{-1} X_n^\top \varepsilon_n\right] \mathbb{E}\left[\varepsilon_n^\top X_n (X_n^\top X_n)^{-1}\right] \\
=\ & \mathbb{E}\left[(X_n^\top X_n)^{-1} X_n^\top \mathbb{E}[\varepsilon_n \varepsilon_n^\top | X_n] X_n (X_n^\top X_n)^{-1}\right] - 0 \\
=\ & \sigma^2 \mathbb{E}(X_n^\top X_n)^{-1}.
\end{aligned}
$$

Since $x \sim N(0, I_d)$, $(X_n^\top X_n)^{-1}$ follows inverse Wishart distribution associated with $(I_d, n)$, and we obtain

$$Var(\widehat{\theta}(0)) = \frac{\sigma^2 I_d}{n - d - 1}.$$

***Case 2,*** $d - n - 1 > 0$*:* when $n < d - n - 1$, following Belkin et al. (2019b), we know that

$$\widehat{\theta}(0) = X_n^\top (X_n X_n^\top)^{-1} Y_n = X_n^\top (X_n X_n^\top)^{-1} X_n \theta_0 + X_n^\top (X_n X_n^\top)^{-1} \varepsilon_n.$$

As a result, the variance matrix of $\widehat{\theta}(0)$ becomes

$$
\begin{aligned}
&Var(\widehat{\theta}(0)) \\
=\ & \mathbb{E}\widehat{\theta}(0)\widehat{\theta}(0)^\top - \mathbb{E}\widehat{\theta}(0)\mathbb{E}\widehat{\theta}(0)^\top \\
=\ & \mathbb{E}X_n^\top (X_n X_n^\top)^{-1} X_n \theta_0 \theta_0^\top X_n^\top (X_n X_n^\top)^{-1} X_n + \mathbb{E}X_n^\top (X_n X_n^\top)^{-1} \varepsilon_n \varepsilon_n^\top (X_n X_n^\top)^{-1} X_n \\
& + \underbrace{\mathbb{E}X_n^\top (X_n X_n^\top)^{-1} X_n \theta_0 \varepsilon_n^\top (X_n X_n^\top)^{-1} X_n}_{=0} \\
& + \underbrace{\mathbb{E}X_n^\top (X_n X_n^\top)^{-1} \varepsilon_n \theta_0 X_n^\top (X_n X_n^\top)^{-1} X_n}_{=0} \\
& - \left[\mathbb{E}X_n^\top (X_n X_n^\top)^{-1} X_n\right] \theta_0 \theta_0^\top \left[\mathbb{E}X_n^\top (X_n X_n^\top)^{-1} X_n\right] \\
=\ & \mathbb{E}X_n^\top (X_n X_n^\top)^{-1} X_n \theta_0 \theta_0^\top X_n^\top (X_n X_n^\top)^{-1} X_n + \sigma^2 \mathbb{E}X_n^\top (X_n X_n^\top)^{-2} X_n \\
& - \left[\mathbb{E}X_n^\top (X_n X_n^\top)^{-1} X_n\right] \theta_0 \theta_0^\top \left[\mathbb{E}X_n^\top (X_n X_n^\top)^{-1} X_n\right].
\end{aligned}
$$

Since each dimension of $x$ is independent $N(0, 1)$, we have

$$tr(\mathbb{E}X_n^\top (X_n X_n^\top)^{-2} X_n) = \mathbb{E}tr((X_n X_n^\top)^{-1}) = \frac{n}{d-n-1}.$$

Using the symmetric property of $X_n X_n^\top$, we have

$$\|\mathbb{E}X_n^\top (X_n X_n^\top)^{-1} X_n\|^2 = \frac{1}{d} tr(\mathbb{E}X_n^\top (X_n X_n^\top)^{-1} X_n) = 1,$$

and all the eigenvalues of $\mathbb{E}X_n^\top (X_n X_n^\top)^{-1} X_n$ are the same, so

$$tr\left(\left[\mathbb{E}X_n^\top (X_n X_n^\top)^{-1} X_n\right] \theta_0 \theta_0^\top \left[\mathbb{E}X_n^\top (X_n X_n^\top)^{-1} X_n\right]\right) = \|\theta_0\|^2.$$

For the trace of $\mathbb{E}X_n^\top (X_n X_n^\top)^{-1} X_n \theta_0 \theta_0^\top X_n^\top (X_n X_n^\top)^{-1} X_n$, we have

$$
\begin{aligned}
&tr\left(\mathbb{E}X_n^\top (X_n X_n^\top)^{-1} X_n \theta_0 \theta_0 X_n^\top (X_n X_n^\top)^{-1} X_n\right) \\
=\ & \mathbb{E}\theta_0^\top X_n^\top (X_n X_n^\top)^{-1} X_n X_n^\top (X_n X_n^\top)^{-1} X_n \theta_0 \\
=\ & \mathbb{E}\theta_0^\top X_n^\top (X_n X_n^\top)^{-1} X_n \theta_0 \\
=\ & \|\theta_0\|^2.
\end{aligned}
$$

As a result,

$$tr(Var(\widehat{\theta}(0))) = \frac{\sigma^2 n}{d-n-1}.$$

$\square$

*Proof of Theorem 1.* We first show that the training trajectory for clean training dominates adversarial training when $\epsilon = o(\sqrt{d/n})$.

Instead of considering the two cases where $\liminf d/n > 0$ or not, we consider three different scenarios w.r.t. $d$: (1) high-dimension regime ($d \gg n$), (2) large-sample regime ($d \ll n$), and (3) moderate dimension regime ($d \asymp n$). These three scenarios are more general in statistical literature.

Denote $X_n \in \mathbb{R}^{n \times d}$ as the data matrix and $Y_n \in \mathbb{R}^n$ as the response vector. Then the updating gradient of square loss in clean training is

$$\frac{\partial \widehat{R}(\theta, 0)}{\partial \theta} = \frac{2}{n}(X_n^\top X_n \theta - X_n^\top Y_n),$$

while the gradient for adversarial training is

$$\frac{\partial \widehat{R}(\theta, \epsilon)}{\partial \theta} = \underbrace{\frac{2}{n}(X_n^\top X_n \theta - X_n^\top Y_n)}_{:=A} + \underbrace{2\epsilon^2 \theta}_{:=B} + \underbrace{\frac{2\epsilon}{n} \frac{\theta}{\|\theta\|} \|X_n \theta - Y_n\|_1}_{:=C} + \underbrace{\frac{2\epsilon \|\theta\|}{n} X_n^\top \operatorname{sgn}(X_n \theta - Y_n)}_{:=D}.$$

- In high-dimension regime, taking $\eta$ small enough so that it is smaller than $1/\lambda_{\max}(X_n^\top X_n)$ ($\lambda_{\max}$ refers to the largest eigenvalue), from the proof of Theorem 5 of Xing et al. (2021b), $\epsilon = o(\sqrt{d/n})$ is a sufficient condition to ensure that $\theta^{(t)}(\epsilon)$ is dominated by $\theta^{(0)}(0)$ for all $t \le T$ with some properly chosen $T$.

- In the large sample regime, one can show that

$$\|\theta^{(t+1)}(\epsilon) - \theta^{(t+1)}(0)\| \le \|\theta^{(t)}(\epsilon) - \theta^{(t)}(0)\| + \eta \left\| \frac{\partial \widehat{R}(\theta^{(t)}(0), 0)}{\partial \theta} - \frac{\partial \widehat{R}(\theta^{(t)}(\epsilon), \epsilon)}{\partial \theta} \right\|,$$

where

$$
\begin{aligned}
& \left\| \frac{\partial \widehat{R}(\theta^{(t)}(0), 0)}{\partial \theta} - \frac{\partial \widehat{R}(\theta^{(t)}(\epsilon), \epsilon)}{\partial \theta} \right\| \\
=\ & \left\| \frac{2}{n} X_n^\top X_n (\theta^{(t)}(0) - \theta^{(t)}(\epsilon)) \right\| + \|B\| + \|C\| + \|D\|.
\end{aligned}
$$

With probability tending to 1, we have for some positive constant $C_1$,

$$\lambda_{\min}\left(\frac{1}{n}X_n^\top X_n - I_d\right) \geq C_1\sqrt{\frac{d}{n}}\frac{1}{\log n}, \quad \lambda_{\max}\left(\frac{1}{n}X_n^\top X_n - I_d\right) \leq C_1\sqrt{\frac{d}{n}}\log n,$$

therefore,

$$\left\|\frac{2}{n}X_n^\top X_n(\theta^{(t)}(0) - \theta^{(t)}(\epsilon))\right\| = \Theta(\|\theta^{(t)}(0) - \theta^{(t)}(\epsilon)\|).$$

On the other hand,

$$\begin{aligned}
\|A\| &= \left\|\frac{2}{n}(X_n^\top X_n\theta - X_n^\top Y_n)\right\| = \frac{2}{n}\sqrt{(X_n\theta - Y_n)^\top(X_n^\top X_n)(X_n\theta - Y_n)}, \\
\|B\| &= 2\epsilon^2\|\theta\|, \\
\|C\| &\leq \frac{2\epsilon}{n}\sqrt{n}\|X_n\theta - Y_n\|, \\
\|D\| &\leq 2\epsilon\|\theta\|\frac{1}{n}\|X_n\|\|\operatorname{sgn}(X_n\theta - Y_n)\|_2 \leq 2\epsilon\|\theta\|\|X_n\|/\sqrt{n}.
\end{aligned}$$

With probability tending to 1, we have $\|A\| \leq C_3\|X_n\theta - Y_n\|/\sqrt{n}$, $\|A\| \geq C_4\|X_n\theta - Y_n\|/\sqrt{n}$, and $\|X_n\| \leq C_5\sqrt{n}$ for some positive constants $C_3, C_4, C_5$. As a result, when $\|\theta - \theta_0\| = \Omega(\sqrt{d/n})$ and $\|X_n\theta - Y_n\|_2^2/n = \Omega(d/n)$, the adversarial training is dominated by $A$.

Rewriting $\theta^{(t)}(\epsilon) = \theta^{(t)}(0) + \theta^{(t)}(\epsilon) - \theta^{(t)}(0)$, we have $\|\theta^{(t)}(\epsilon)\| \leq \|\theta^{(t)}(0)\| + \|\theta^{(t)}(\epsilon) - \theta^{(t)}(0)\|$. Similarly we have $\|X_n\theta^{(t)}(\epsilon) - Y_n\| \leq \|X_n\theta^{(t)}(0) - Y_n\| + \|X_n(\theta^{(t)}(\epsilon) - \theta^{(t)}(0))\|$, and for some positive constants $C_6$ to $C_{10}$

$$\begin{aligned}
\|\theta^{(t+1)}(\epsilon) - \theta^{(t+1)}(0)\| &\leq (1 + C_6\eta)\|\theta^{(t)}(\epsilon) - \theta^{(t)}(0)\| + \eta\|B\| + \eta\|C\| + \eta\|D\| \\
&\leq (1 + C_6\eta + C_7\epsilon\eta)\|\theta^{(t)}(\epsilon) - \theta^{(t)}(0)\| + C_8\epsilon\eta\|\theta^{(t)}(0)\| \\
&\quad + C_9\epsilon\eta\|X_n\theta^{(t)}(0) - Y_n\|/\sqrt{n}.
\end{aligned}$$

Based on Ba et al. (2020), we have with probability tending to 1, $\|\theta^{(t)}(0)\|$ and $\|X_n\theta^{(t)}(0) - Y_n\|/\sqrt{n}$ are both bounded, thus we have

$$\|\theta^{(t+1)}(\epsilon) - \theta^{(t+1)}(0)\| \leq (1 + C_{10}\eta)\|\theta^{(t)}(\epsilon) - \theta^{(t)}(0)\| + C_{11}\epsilon\eta.$$

Taking $(\eta, T)$ such that $\eta \to 0$, $T = O(\operatorname{poly}(d, n))$, $\eta T \to \infty$, $\eta T/(\log n) \to 0$, we have for all $t \leq T$,

$$\|\theta^{(t)}(\epsilon) - \theta^{(t)}(0)\| = (1 + \eta t)\epsilon = o(\sqrt{d/n}).$$

In addition, based on Xing et al. (2021b), we also have $\theta^{(T)}(\epsilon) \to \widehat{\theta}(\epsilon)$ in this case.

- For moderate dimension case, the proof is similar.

When $\epsilon \gg \sqrt{d/n}$,

- When $d/n < \infty$, the optimum solutions for $\widehat{R}(\theta, \epsilon)$ and $\widehat{R}(\theta, 0)$ are already different, so their training trajectory are different as well.

- When $d/n \to \infty$, we have both $\|\widehat{\theta}(\epsilon)\| \to 0$ and $\|\widehat{\theta}(0)\| \to 0$. However, in clean training, there is a stage that $\|\theta^{(t)}(0)\| = O(\sqrt{n/d})$ while $\|X_n\theta^{(t)}(0) - Y_n\|^2/n \to 0$, so $B$ will finally dominates $\partial\widehat{R}(\theta, \epsilon)/\partial\theta$.

$\square$

## C.2 Theorem 2

**Lemma 2.** *Under the conditions in Theorem 1,*

- *When $d/n < \infty$,*
$$\Delta = O_p(\sqrt{d/n}).$$

- *When $d/n \to \infty$,*
$$\Delta = O_p(1).$$

*Proof of Lemma 2.* Since $\epsilon = 0$, we can obtain the closed-form solution of $\widehat{\theta}(0)$.

**Case 1,** $n - d - 1 > 0$, we know that
$$\widehat{\theta}(0) = \theta_0 + (X_n^\top X_n)^{-1} X_n^\top \varepsilon_n,$$

therefore,
$$
\begin{aligned}
\Delta &= \mathbb{E}_{(X,\varepsilon)}(X(X_n^\top X_n)^{-1} X_n^\top \varepsilon_n - \varepsilon)^2 - \|X_n(X_n^\top X_n)^{-1} X_n^\top \varepsilon_n - \varepsilon_n\|^2/n \\
&= \varepsilon_n^\top X_n (X_n^\top X_n)^{-2} X_n^\top \varepsilon_n + \sigma^2 - \|X_n(X_n^\top X_n)^{-1} X_n^\top \varepsilon_n - \varepsilon_n\|^2/n,
\end{aligned}
$$

where
$$
\mathbb{E}\|X_n(X_n^\top X_n)^{-1} X_n^\top \varepsilon_n - \varepsilon_n\|^2/n = \frac{\sigma^2}{n}\mathbb{E}tr\left(I_n - X_n(X_n^\top X_n)^{-1} X_n\right) = \frac{\sigma^2(n-d)}{n},
$$

and
$$
\begin{aligned}
&\mathbb{E}\|X_n(X_n^\top X_n)^{-1} X_n^\top \varepsilon_n - \varepsilon_n\|^4/n^2 \\
&= 2\sigma^4 \mathbb{E}tr(I_n - X_n(X_n^\top X_n)^{-1} X_n) + \mathbb{E}^2\|X_n(X_n^\top X_n)^{-1} X_n^\top \varepsilon_n - \varepsilon_n\|^2/n \\
&= 2\sigma^4 \frac{n-d}{n} + \sigma^4 \frac{(n-d)^2}{n^2}.
\end{aligned}
$$

As a result, the mean and variance of $\Delta$ become
$$
\begin{aligned}
\mathbb{E}\Delta &= \sigma^2\left(tr(\mathbb{E}(X_n^\top X_n)^{-1}) + 1\right) - \mathbb{E}\|(X_n^\top X_n)^{-1} X_n \varepsilon_n - \varepsilon_n\|^2/n \\
&= \sigma^2 \frac{d}{n-d-1} + \sigma^2 - \sigma^2 \frac{n-d}{n} \\
&= \sigma^2 d\left(\frac{1}{n-d-1} - \frac{1}{n}\right),
\end{aligned}
$$

and
$$
\begin{aligned}
\mathbb{E}\Delta^2 &= \mathbb{E}\left(\varepsilon_n^\top X_n(X_n^\top X_n)^{-2} X_n^\top \varepsilon_n + \sigma^2 - \|X_n(X_n^\top X_n)^{-1} X_n^\top \varepsilon_n - \varepsilon_n\|^2/n\right)^2 \\
&= \mathbb{E}\left(\varepsilon_n^\top X_n(X_n^\top X_n)^{-2} X_n^\top \varepsilon_n + \sigma^2\right)^2 + \mathbb{E}\|X_n(X_n^\top X_n)^{-1} X_n^\top \varepsilon_n - \varepsilon_n\|^4/n^2 \\
&\quad - 2\mathbb{E}\left(\varepsilon_n^\top X_n(X_n^\top X_n)^{-2} X_n^\top \varepsilon_n + \sigma^2\right)\|X_n(X_n^\top X_n)^{-1} X_n^\top \varepsilon_n - \varepsilon_n\|^2/n,
\end{aligned}
$$

where
$$
\begin{aligned}
&\mathbb{E}\left(\varepsilon_n^\top X_n(X_n^\top X_n)^{-2} X_n^\top \varepsilon_n + \sigma^2\right)^2 \\
&= 2\sigma^4 \mathbb{E}tr\left(X_n(X_n^\top X_n)^{-3} X_n^\top\right) + \sigma^4 \frac{d^2}{(n-d-1)^2} + \sigma^4 + 2\sigma^4 \frac{d}{n-d-1},
\end{aligned}
$$

$$
\begin{aligned}
&\mathbb{E}\left(\varepsilon_n^\top X_n(X_n^\top X_n)^{-2} X_n^\top \varepsilon_n + \sigma^2\right)\|X_n(X_n^\top X_n)^{-1} X_n^\top \varepsilon_n - \varepsilon_n\|^2/n \\
&= 2\sigma^4 \frac{1}{n}\mathbb{E}tr\left(X_n(X_n^\top X_n)^{-2} X_n^\top\right) + \sigma^4 \frac{d}{n-d-1}\frac{n-d}{n} + \sigma^4 \frac{n-d}{n} \\
&= 2\sigma^4 \frac{d}{n-d-1} + \sigma^4 \frac{d}{n-d-1}\frac{n-d}{n} + \sigma^4 \frac{n-d}{n}.
\end{aligned}
$$

One can conclude that $\mathbb{E}\Delta^2 = \Theta(d/n)$ and thus $\Delta = O_p(\sqrt{d/n})$.

**Case 2,** $n/d \to 0$, we know that $\widehat{R}(\widehat{\theta}(0), 0) = 0$ due to data interpolation, and $\mathbb{E}\|\widehat{\theta}(0)\|^2 = \Theta(n/d)$, thus one can directly obtain that $\Delta = O_p(1)$.

**Case 3,** $d - n - 1 > 0$ *while $n/d > 0$, the analysis is similar to Case 1.*  $\qquad\square$

*Proof of Theorem 2.* When $d/n < \infty$, we have with probability tending to 1,

$$
\begin{aligned}
& \widehat{R}(\widehat{\theta}(0), 0) + O(\epsilon + \epsilon^2) \\
\leq \; & \widehat{R}(\widehat{\theta}(\epsilon), 0) + O(\epsilon + \epsilon^2) \\
\leq \; & \widehat{R}(\widehat{\theta}(\epsilon), \epsilon) \\
\leq \; & \widehat{R}(\widehat{\theta}(0), \epsilon) \\
= \; & \widehat{R}(\widehat{\theta}(0), 0) + O(\epsilon + \epsilon^2),
\end{aligned}
$$

which means that $\epsilon = \Theta(\sqrt{d/n})$ is the threshold of $(\widehat{R}(\widehat{\theta}(\epsilon), \epsilon) - \widehat{R}(\widehat{\theta}(0), 0))/\Delta$.

When $d/n \to \infty$, if $\epsilon = o(\sqrt{d/n})$, we have

$$
\widehat{R}(\widehat{\theta}(0), \epsilon) = O(\epsilon^2 \|\widehat{\theta}(0)\|^2) = o(1).
$$

Thus

$$
\widehat{R}(\widehat{\theta}(\epsilon), \epsilon) \leq \widehat{R}(\widehat{\theta}(0), \epsilon) = o(1),
$$

and

$$
(\widehat{R}(\widehat{\theta}(\epsilon), \epsilon) - \widehat{R}(\widehat{\theta}(0), 0))/\Delta \to 0.
$$

When $d/n \to \infty$ and $\epsilon \gg \sqrt{d/n}$, $\widehat{R}(\theta, \epsilon)$ is dominated by $\epsilon^2 \|\theta\|^2$, so $\|\widehat{\theta}(\epsilon)\|$ is small enough so that $\epsilon^2 \|\widehat{\theta}(\epsilon)\|^2 = O(1)$. When $\|\theta\|$ is small enough, we have $\widehat{R}(\theta, 0) = \Theta(1)$ in probability. These observations imply that $\widehat{R}(\widehat{\theta}(\epsilon), \epsilon) = \Theta(1)$ in probability. Consequently, $(\widehat{R}(\widehat{\theta}(\epsilon), \epsilon) - \widehat{R}(\widehat{\theta}(0), 0))/\Delta$ does not converge to zero.

$\square$

## C.3 Proposition 1

The proof idea is the same as Ba et al. (2020) and Xing et al. (2021b). When $h \to 0$, for smooth activation functions, one can use first-order Taylor expansion to approximate them as linear functions. As a result, with vanishing initialization, the two-layer neural network is approximately a linear network (a linear function), thus the results follows Theorem 1 and 2.

In particular, we assume

- The activation function $\phi$ in model (3) is twice continuously differentiable, $\phi'(0) \neq 0$, and $\phi(0) = 0$.
- The initialization satisfies $\theta_{\xi,j}^{(0)} \sim N(0, I_d/dh^{1+\delta})$ for some $\delta > 0$.
- The parameter $a$ satisfy $\|a\|_\infty = O(1)$, $\max |a_j|/(\min |a_j|) = \Theta(1)$.

Then when $h$ is sufficiently large and $\delta$ is large enough, taking $\eta = \eta_{linear} h/(\|a\|^2 \phi'(0)^2)$ and suitable $T \to \infty$, we have with probability tending to 1 over the generation of $x$,

$$
\left| \frac{1}{\sqrt{h}} \sum_{j=1}^h \phi(x^\top \theta_j^{(T)}) a_j - x^\top \theta_{linear}^{(T)} \right| = o(\sqrt{d/n} \wedge \sqrt{n/d}),
$$

where $\eta_{linear}$ and $\theta_{linear}^{(T)}$ are the learning rate and model parameters corresponding to the linear model.

## C.4 Proposition 2

*Proof of Proposition 2.* Denote $g_t = \partial \widehat{R}(\theta^{(t)}, \epsilon)/\partial \theta^{(t)}$. The updating rule leads to

$$
\|\theta^{(t)} - \widehat{\theta}\|^2 \leq \|\theta^{(t-1)} - \widehat{\theta} - \eta_t g_t\|^2 \leq \|\theta^{(t-1)} - \widehat{\theta}\|^2 - 2\eta_t g_t^\top (\theta^{(t-1)} - \widehat{\theta}) + \eta_t^2 L^2.
$$

Taking expectation and move some terms, it becomes

$$g_t^\top(\theta^{(t-1)} - \widehat{\theta}) \le \frac{1}{2\eta_t}\|\theta^{(t-1)} - \widehat{\theta}\|^2 - \frac{1}{2\eta_t}\|\theta^{(t)} - \widehat{\theta}\|^2 + \frac{1}{2}\eta_t L_r^2.$$

Taking average over $t = 1$ to $T$, we have

$$
\begin{aligned}
\frac{1}{T}\left[\sum_{t=1}^T g_t^\top(\theta^{(t-1)} - \widehat{\theta})\right] &\le \frac{1}{2T}\left[\sum_{t=1}^T \frac{1}{\eta_t}\|\theta^{(t-1)} - \widehat{\theta}\|^2 - \frac{1}{\eta_t}\|\theta^{(t)} - \widehat{\theta}\|^2\right] + \frac{L^2}{2T}\sum_{t=1}^T \eta_t \\
&= \frac{\|\theta^{(0)} - \widehat{\theta}\|^2}{2\eta_1 T} - \frac{\|\theta^{(T)} - \widehat{\theta}\|^2}{2\eta_T T} \\
&\quad + \frac{1}{2T}\left[\sum_{t=1}^{T-1}\left(\frac{1}{\theta_{t+1}} - \frac{1}{\theta_t}\right)\|\theta^{(t)} - \widehat{\theta}\|^2\right] + \frac{L^2}{2T}\sum_{t=1}^T \eta_t.
\end{aligned}
$$

Finally, since $\widehat{R}$ is a convex function, we have

$$\frac{1}{T}\left[\sum_{t=1}^T g_t^\top(\theta^{(t-1)} - \widehat{\theta})\right] \ge \frac{1}{T}\sum_{t=1}^T \widehat{R}(\theta^{(t)}) - \widehat{R}(\widehat{\theta}) \ge \left[\min_{t=1,\dots,T} \widehat{R}(\theta^{(t)}) - \widehat{R}(\widehat{\theta})\right].$$

Further, to show that $\widehat{\epsilon} - \epsilon^* = o(\epsilon^*)$, it suffices to show that $\widehat{R}(\widehat{\theta}(\epsilon), \epsilon)$ is differentiable and has finite second order derivative when $\epsilon = o(1)$.

Recall the first order optimality condition for $\widehat{R}(\theta, \epsilon)$ is

$$\frac{\partial \widehat{R}(\widehat{\theta}(\epsilon), \epsilon)}{\partial \theta} = \mathbf{0}.$$

Therefore, take $\Delta\epsilon \to 0$,

$$
\begin{aligned}
\mathbf{0} &= \frac{\partial \widehat{R}(\widehat{\theta}(\epsilon), \epsilon)}{\partial \theta} - \frac{\partial \widehat{R}(\widehat{\theta}(\epsilon + \Delta\epsilon), \epsilon + \Delta\epsilon)}{\partial \theta} \\
&= \frac{2}{n}(X_n^\top X_n \widehat{\theta}(\epsilon) - X_n^\top Y_n) + 2\epsilon^2 \widehat{\theta}(\epsilon) + \frac{2\epsilon}{n}\frac{\widehat{\theta}(\epsilon)}{\|\widehat{\theta}(\epsilon)\|}\|X_n\widehat{\theta}(\epsilon) - Y_n\|_1 \\
&\quad + \frac{2\epsilon\|\widehat{\theta}(\epsilon)\|}{n} X_n^\top \operatorname{sgn}(X_n\widehat{\theta}(\epsilon) - Y_n) \\
&\quad - \frac{2}{n}(X_n^\top X_n \widehat{\theta}(\epsilon + \Delta\epsilon) - X_n^\top Y_n) - 2(\epsilon + \Delta\epsilon)^2 \widehat{\theta}(\epsilon + \Delta\epsilon) \\
&\quad - \frac{2(\epsilon + \Delta\epsilon)}{n}\frac{\widehat{\theta}(\epsilon + \Delta\epsilon)}{\|\widehat{\theta}(\epsilon + \Delta\epsilon)\|}\|X_n\widehat{\theta}(\epsilon + \Delta\epsilon) - Y_n\|_1 \\
&\quad - \frac{2(\epsilon + \Delta\epsilon)\|\widehat{\theta}(\epsilon + \Delta\epsilon)\|}{n} X_n^\top \operatorname{sgn}(X_n\widehat{\theta}(\epsilon + \Delta\epsilon) - Y_n) \\
&= \frac{2}{n}X_n^\top X_n(\widehat{\theta}(\epsilon) - \widehat{\theta}(\epsilon + \Delta\epsilon)) + 2\epsilon^2(\widehat{\theta}(\epsilon) - \widehat{\theta}(\epsilon + \Delta\epsilon)) + 2\epsilon\Delta\epsilon\widehat{\theta}(\epsilon) \\
&\quad + \frac{2\Delta\epsilon}{n}\frac{\widehat{\theta}(\epsilon)}{\|\widehat{\theta}(\epsilon)\|}\|X_n\widehat{\theta}(\epsilon) - Y_n\|_1 \\
&\quad + \frac{2\epsilon}{n}\left[\frac{\widehat{\theta}(\epsilon)}{\|\widehat{\theta}(\epsilon)\|}\|X_n\widehat{\theta}(\epsilon) - Y_n\|_1 - \frac{\widehat{\theta}(\epsilon + \Delta\epsilon)}{\|\widehat{\theta}(\epsilon + \Delta\epsilon)\|}\|X_n\widehat{\theta}(\epsilon + \Delta\epsilon) - Y_n\|_1\right] \\
&\quad + \frac{2\Delta\epsilon}{n}\|\widehat{\theta}(\epsilon)\| X_n^\top \operatorname{sgn}(X_n\widehat{\theta}(\epsilon) - Y_n) \\
&\quad + \frac{2\epsilon}{n}\left[\|\widehat{\theta}(\epsilon)\| X_n^\top \operatorname{sgn}(X_n\widehat{\theta}(\epsilon) - Y_n) - \|\widehat{\theta}(\epsilon + \Delta\epsilon)\| X_n^\top \operatorname{sgn}(X_n\widehat{\theta}(\epsilon + \Delta\epsilon) - Y_n)\right] + o.
\end{aligned}
$$

As a result, putting $(\widehat{\theta}(\epsilon) - \widehat{\theta}(\epsilon + \Delta\epsilon))$ on one side and $\Delta\epsilon$ on the other side of the above equation, we have

$$
\left[ \frac{1}{n} X_n^\top X_n + \epsilon^2 + \frac{\epsilon}{n} \frac{\partial}{\partial\theta} \left( \frac{\widehat{\theta}(\epsilon)}{\|\widehat{\theta}(\epsilon)\|} \|X_n\widehat{\theta}(\epsilon) - Y_n\|_1 \right) + \frac{\epsilon}{n} \frac{\partial\|\widehat{\theta}(\epsilon)\| X_n^\top \operatorname{sgn}\left(X_n\widehat{\theta}(\epsilon) - Y_n\right)}{\partial\theta} \right]
$$

$$
\times (\widehat{\theta}(\epsilon) - \widehat{\theta}(\epsilon + \Delta\epsilon))
$$

$$
= \Delta\epsilon \left[ \epsilon\widehat{\theta}(\epsilon) + \frac{1}{n} \frac{\widehat{\theta}(\epsilon)}{\|\widehat{\theta}(\epsilon)\|} \|X_n\widehat{\theta}(\epsilon) - Y_n\|_1 + \frac{1}{n} \|\widehat{\theta}(\epsilon)\| X_n^\top \operatorname{sgn}\left(X_n\widehat{\theta}(\epsilon) - Y_n\right) + o \right],
$$

which implies that

$$
\frac{\widehat{\theta}(\epsilon) - \widehat{\theta}(\epsilon + \Delta\epsilon)}{\Delta\epsilon}
$$

$$
= \left[ \frac{1}{n} X_n^\top X_n + \epsilon^2 + \frac{\epsilon}{n} \frac{\partial}{\partial\theta} \left( \frac{\widehat{\theta}(\epsilon)}{\|\widehat{\theta}(\epsilon)\|} \|X_n\widehat{\theta}(\epsilon) - Y_n\|_1 \right) \right.
$$

$$
\left. + \frac{\epsilon}{n} \frac{\partial\|\widehat{\theta}(\epsilon)\| X_n^\top \operatorname{sgn}\left(X_n\widehat{\theta}(\epsilon) - Y_n\right)}{\partial\theta} \right]^{-1}
$$

$$
\times \left[ \epsilon\widehat{\theta}(\epsilon) + \frac{1}{n} \frac{\widehat{\theta}(\epsilon)}{\|\widehat{\theta}(\epsilon)\|} \|X_n\widehat{\theta}(\epsilon) - Y_n\|_1 + \frac{1}{n} \|\widehat{\theta}(\epsilon)\| X_n^\top \operatorname{sgn}\left(X_n\widehat{\theta}(\epsilon) - Y_n\right) + o \right].
$$

Note that due to the distribution of $X_n$ and $Y_n$, $\|X_n\widehat{\theta}(\epsilon) - Y_n\|_1/n$ and $\operatorname{sgn}\left(X_n\widehat{\theta}(\epsilon) - Y_n\right)/n$ are approximately smooth. In addition, since both $\epsilon^*$ and $\epsilon$ are $o(1)$, this corresponds to the large-sample and moderate-dimension regimes, thus $\|\widehat{\theta}(\epsilon)\|$ diverges from zero, and $\frac{\partial}{\partial\theta} \left( \frac{\widehat{\theta}(\epsilon)}{\|\widehat{\theta}(\epsilon)\|} \|X_n\widehat{\theta}(\epsilon) - Y_n\|_1 \right)$ exists as well.

Using similar technique, one can obtain the second-order derivative of $\widehat{R}(\widehat{\theta}(\epsilon), \epsilon)$ w.r.t. $\epsilon$. $\qquad\square$