# OpenReview forum: "Phase Transition from Clean Training to Adversarial Training"
_NeurIPS.cc/2022/Conference — NeurIPS 2022 Accept_

### Official Review · Reviewer_T2si · 2022-07-01

**Rating:** 3
**Confidence:** 3
**Soundness:** 2 fair
**Presentation:** 3 good
**Contribution:** 2 fair

**Summary:**

The authors study the behavior of a simple linear model (and a shallow NN) under adversarial training and theoretically discover qualitative differences between the small-$\epsilon$ regime and the large-$\epsilon$ regime. They claim that these regimes are separated by a critical perturbation strength $\epsilon^*$ and propose a theoretically motivated method of determining it. They run experiments on both their simple models as well as standard ResNets in order to validate their theoretical findings.

**Questions:**

In Theorem 1, it appears that the $\epsilon$ refers not only to the attack strength but actually to the true Gaussian noise present in the data. Why does it make sense to jointly vary the true noise on the data and the budget of the adversarial training? Is the $\epsilon$ even referring to the adversarial attack strength at all here?

In line 208, the authors talk about running adversarial training for a wide range of $\epsilon$'s. Are they referring to the results in Table 1? Because, if so, that is not a wide range of $\epsilon$'s. If not, then more results should be shown, ideally as plots.

**Limitations:**

The authors do not discuss societal impact of their work, but I agree with their assessment in the checklist that this does not apply to their work, as it is fairly theoretical.

**Strengths And Weaknesses:**

he paper is well-structured and given that the paper is technically complicated in some parts, the authors do a good job of clearly conveying their message.

A central claim of the paper is the existence of a phase transition when increasing $\epsilon$ through some critical value. Usually, phase transitions are characterized by a discontinuity in one or more properties of the system. To me it seems that (at fixed dimension $d$ and samples $n$) the paper simply shows that $\epsilon=0$ and $\epsilon=\infty$ differ in loss and in optimal solution. This is completely obvious. What would be interesting is if some property of the system actually varies non-smoothly as we pass through a critical value of $\epsilon$. The most direct way of studying this is by varying $\epsilon$ in a fine grained way and demonstrating a discontinuity in one or several properties of the network. The authors justify why they do not do this by the high cost of running adversarial training (if this is a problem, the authors could consider using the method of [1] for their experiments).

As it is, I do not think that the authors provide enough evidence of an interesting discontinuity. For example, on the right-hand side of Figure 2, it seems fairly clear that the loss distribution is varying smoothly as a function of $\epsilon$. This is not a surprising result. A similar observation is true for Figure 3 and an even more striking counter-example to the authors claim of a phase transition seems to be their own Figure A.1 in the appendix. The only weak evidence that I think the paper provides for the phase transition claim is Figure 4: here, looking at the generalization gap ratio after overfitting at 200 epochs, it seems that there might be a non-smooth increase as we vary $\epsilon$.

[1] "Fast is better than free: Revisiting adversarial training" Wong, Rice, Kolter at ICLR'20

---

> ### Author Response · Authors · 2022-08-02
> **Response**
>
> Thank you for reviewing our paper! Below are some responses to your questions and comments:
>
> 1. Q1, $\varepsilon$ and $\epsilon$: These two are different notations. The former notation $\varepsilon$ is the noise term in the response $y$, which has nothing to do with the attack. The latter notation $\epsilon$ denotes the strength of the adversarial attack.
>
> 2. Q2, Table1: Although we ran experiments for various $\epsilon$, our aim is to find $\epsilon^*$. As a result, we only present several trials where $\epsilon$ is around  $\epsilon^*$(=0.5).
>
> 3. Weakness:
>
>     * Our paper is not trying to show that $\epsilon=0$ and $\epsilon=\infty$ lead to different adversarial training trajectories. We aim to justify that the two cases, $\lim \epsilon/\epsilon^*=0$ or $\lim\epsilon/\epsilon^*=\infty$, have different adversarial training trajectories, where we consider asymptotically $d$ and $n$ increase toward infinite.
>
>     * Since our phase transition is an asymptotic order instead of a critical value (e.g., the melting point of ice), under the finite sample setting, the transition will be smooth when increasing $\epsilon$ smoothly. Therefore, we take $\epsilon$ on the two sides of $\epsilon^*$ and make them as large/small as possible representing a $\epsilon$ value that is larger or smaller than $\epsilon^*$ in asymptotic order. To make our numerical results more convincing, we prepare to add more simulation results for a grid value of $\epsilon$ around $\epsilon^*$, aiming to show that the change of adversarial training behavior (which can be characterized via p-value of hypothesis testing) starts to explode at $\epsilon^*$.

---

### Official Review · Reviewer_Xy6A · 2022-07-08

**Rating:** 4
**Confidence:** 3
**Soundness:** 2 fair
**Presentation:** 3 good
**Contribution:** 3 good

**Summary:**

Adversarial training does not achieve performance comparable to the standard training on clean data. This paper investigates the critical point of the magnitude of the adversarial perturbations with which training trajectories of adversarial training become significantly different from those of standard training. By using simple linear regression models, this paper claims that the order of the critical point is in $\Theta(\sqrt{d/n})$ where $d$ is the dimension of data, and $n$ is data size.
In addition, it claims that the critical point can be estimated by using the difference between training loss of adversarial training and test loss of standard training. To support the claims, this paper provides results of various experiments, e.g., evaluation of the difference of parameters in standard training and adversarial training in the cases (d<<n and n<<d), connectivity of the loss, and empirical evaluation of the critical point.


**Questions:**

What do lim inf d/n > 0 and lim d/n=0 in Theorem 1 mean? I think d and n are fixed in training.

**Limitations:**

Theoretical results are restricted to simple linear models and specific two-layer networks.  Though I admit that deriving theoretical results for deep neural networks are difficult due to nonlinearity, experiments might be made more sophisticated to convince that theoretical results are valid for deep models. For example, $\epsilon$ in Table 1 could be evaluated at finer intervals to show the phase transition experimentally.

**Strengths And Weaknesses:**

## Strengths
- This paper provides interesting analyses of training trajectories of adversarial training.
This study will help readers better understand adversarial training and may inspire the creation of new and stronger methods for adversarial training.

- This paper claims the relationship between the critical point and catastrophic overfitting of FGSM.
If we know the magnitude of attacks against which FGSM can make models sufficiently robust, it is worth in practice because FGSM is an efficient method. However, the results of the experiments (Table 2) are less informative because they do not list the robust accuracy of FGSM against PGD.

## Weaknesses
- The theoretical analysis is based on simple models. Though I admit the analysis for deep models is difficult, experimental evidence of deep models can be obtained instead of theoretical results. In the current paper, the experimental results are not sufficient as described below.

- The experimental conditions are not complete, and the results do not fully support the claim. The simulation on the left of Figure 2 provides little information because $\varepsilon$ varies across conditions of n<<d and d<<n. I think that if $d$ is fixed across the conditions, small and large $\varepsilon$ can be fixed across conditions to compare the difference between n<<d and d<<n conditions on the same scale. It would be useful if these results support Theorem 1. In addition, the experimental conditions on the right side of Figure 2 are not shown. It seems to be important to know what the value of $\sqrt{n/d}$ is in this setting.

- If this paper claims a phase transition, it is not convincing unless experiments show the discontinuous change in the trajectory against $\varepsilon$. In Table 1, the evaluation interval of $\varepsilon$ is too coarse and the losses seem to increase almost linearly, and there is no discontinuity like a phase transition. In addition, Experimental conditions are chosen artificially. For example, Figure 4 shows the results of $\varepsilon$=1/255, 4/255, but the results of $\varepsilon$=1/255, 4/255 are not in Table 1 and Figure 3. Conversely, the results of $\varepsilon$=2/255 are shown in Table 1 and Figure 3 but not in Figure 4. These discrepancies in experimental conditions could be seen as cherry-picking in order to claim that the critical point is $\varepsilon=0.5/255$. In my opinion, many experiments should be conducted on the interval [0,0.5/255] and on the interval [0.5/255,2/255] to obtain more convincing results.

- The proof of theoretical results is unclear, and its validity is difficult to evaluate.  Since the claims in theorems are different from the claims in the proof, it is difficult to follow the proof. For example, Theorem 1 uses lim inf d/n > 0 and lim d/n=0 in the claims, but its proof uses $d/n<\infty$ and $d/n\rightarrow \infty$, and the relationship is not explained. In line 635, A, B, C, D should be written by using $\hat{\theta}(\varepsilon)$. To improve clarity, theorems and proofs should be written in a consistent and easy-to-understand notation.

## Minor issues and Comments.
- In Table 1, $\varepsilon$ is divided by 255, but in the text, $\varepsilon$ is 0.5 without being divided by 255. It is necessary to rewrite the text to determine which is correct.

- In my opinion, Proposition 1 is incomplete. This theoretical claim contains unclearness though it is supplemented by the footnote. I think the term "the similar property but not the same" should be explained exactly in this proposition.

- [a] reports that adversarial training can improve the accuracy on clean data under certain conditions. If this paper can connect such studies to the study fo the difficulty of adversarial training, this paper might be more valuable. For example, if the paper can reveal how weak attacks contribute to clean accuracy, this paper would be worth publishing.

[a] Xie, Cihang, et al. "Adversarial examples improve image recognition." CVPR2020.

---

> ### Author Response · Authors · 2022-08-02
> **Response**
>
> Thank you for reviewing our paper! Below are some answers for your questions:
>
> 1. Weakness, simulation: We appreciate you sharing this comment with us! We will add some more experiments to show the existence of the phase transition, as described later.
>
> 2. Weakness, the choice of $\epsilon$ in different experiments: Yes, we change the choice of $\epsilon$ in different experiments.
>
>     * For Table 1, we sequentially try $\epsilon=0, 0.5, 1, \dots$, and find that $\epsilon^*=0.5$, thus we stop the further searching.
>
>     * For Figures 3 and 4, since our phase transition boundary is an asymptotic order instead of a critical value (e.g., the melting point of ice), we take $\epsilon$ on the two sides of $\epsilon^*$ and make them as large/small as possible to represent a $\epsilon$ value that is larger or smaller than $\epsilon^*$ in asymptotic order. And also, for this reason, Figures 3 and 4 take different choices of $\epsilon$ than Table 1.
>
>     * In terms of the difference between Figure 3 and Figure 4, it is just because Figure 4 already includes too many choices of $\epsilon$. We leave the very large/small $\epsilon$ and pick some other moderate $\epsilon$ so that Figure 4 is clean.
>
>     * On the other hand, we prepare to add more simulation results for a grid value of $\epsilon$ around $\epsilon^*$, aiming to show that the change of adversarial training behavior (which can be characterized via p-value of hypothesis testing) starts to explode at $\epsilon^*$.
>
> 3. Weakness, proof: Thank you for pointing this out! We are providing stronger proof in the appendix than what is used in the paper. We will consider rephrasing the appendix so that it is more clear to check the proofs.
>
> 4. Question, fix $d$ and $n$: In our paper, we are considering asymptotic changes in $d$ and $n$.

---

### Official Review · Reviewer_rFja · 2022-07-09

**Rating:** 6
**Confidence:** 4
**Soundness:** 3 good
**Presentation:** 2 fair
**Contribution:** 3 good

**Summary:**

This paper propose a way to study the effects of the strength of adversarial training on the trajectories of optimization of machine learning models. The specific examples used in this paper is deep neural networks. The proven theorems are on a much simpler linear model.

**Questions:**

no questions

**Ethics Review Area:**

["I don’t know"]

**Limitations:**

The authors have not discuss potential negative societal impact.

**Strengths And Weaknesses:**

Weakness:
1. The phrase "phase transition" must be use with care. In the theory of critical phenomena, phase transitions means existence of singularities in the thermodynamics limit. Most common types of phase transitions are first order and second order phase transitions. I recommend the authors to use another descriptor other than 'phase transition'.
2. The paper prove theorems for simple linear model, I am skeptical that these results can be generalised to arbitrary models.
3. The key equation in the paper is the definition of \epsilon^* in line 151. I cannot make the connection of the theorems with this statement in line 150: "This equivalency justifies our idea of . . . . ". I can hardly understand what was being justified. I feel that definition of \epsilon^* may be intuitive.
4. Is there a typo in line 132? "lim inf" means what? Also, for taking limits, it is better to state what quantity tends to this limit.
5. Line 216: about connectivity. this analysis may not bring too much value in the understanding of the content of this paper.
6. Table 2 is not being explained well.

Strengths:
1. While many past papers focus on prediction accuracies, this paper focus on understanding of the training process. Indeed the focus on tuning parameters and network to make predictions just that little bit better may not help this research field move forward. Papers focusing on understanding brings more value to this field of research. Any paper that push the boundaries of understanding should be encouraged.

---

> ### Author Response · Authors · 2022-08-02
> **Response**
>
> We appreciate your effort in reviewing our paper! Below are the responses for particular questions:
>
> 1. Weakness, Q1, naming: Thanks for pointing out your concern. Our word ``transition" is used in an asymptotic way, i.e., when $d/n$ is larger/smaller than some rate, then the phenomenons will be very different in the two regimes. This is different from the common sense of phase transition in physics, e.g., the melting point of ice.
>
> 2. Weakness, Q2, generalize to other models: We agree that it is hard to extend the theoretical analysis to general models, but we expect that the insight we obtained from simpler model generalizes well. Therefore, we conduct plenty of experiments in neural networks to justify that the phase transition phenomenon also happens in neural networks.
>
> 3. Weakness, Q3, ``equivalence": Thank you for pointing out this unclear expression! Our goal is to show that Theorems 1 and 2 lead to the same critical threshold for $\epsilon$, rather than saying they are mathematically equivalent. We adjusted the word we used in line 150-151 in the revision.
>
> 4. Weakness, Q4, $\lim\inf$: $\lim\inf$ means ``limit inferior" for a sequence of numbers. Its formal definition is $\lim\inf x_n = \lim_{n\rightarrow\infty}(\inf_{m\geq n}x_m)$. It is possible that the sequence of numbers $x_n$ does not have a unique limiting point. In this case,  $\lim\inf$ gives the smallest limiting point.
>
> 5. Weakness, Q5, connectivity: Thanks for sharing your understanding with us! We put connectivity analysis in this paper because it is an important property in neural networks. The whole Section 5 aims to show the various properties in the neural networks change significantly when adversarial training uses a large attack strength $\epsilon$.
>
> 6. Weakness, Q6, Table 2: Thanks for figuring this out! We add more explanations for Table 2 in the revision (line 264-268).

---

### Official Review · Reviewer_ZEbX · 2022-07-18

**Rating:** 4
**Confidence:** 3
**Soundness:** 3 good
**Presentation:** 3 good
**Contribution:** 2 fair

**Summary:**

The paper studies how to mitigate the gap between adversarial training and clean training by finding the optimal $\epsilon$. They first validate the phase transition boundary in simple linear regression model then extend to large-scale neural network. Based on their observation, the commonly used attack strengths are greater than optimal $\epsilon$. They proposed an efficient way to approximate the $\epsilon^\star$ which can make the adversarial training more reliable and evaluate on three dataset.

**Questions:**

The author claims the proposed method can find an optimal $\epsilon$ for adversarial training. I'm wondering if there has a tradeoff between robust acc. and clean acc.. Table 1. shows the loss and acc. value between training and testing, but the author only focus on the explanation of loss value.

Can it be used to evaluate on the ensemble attack method such as  AutoAttack?

**Limitations:**

Yes

**Strengths And Weaknesses:**

Strength: the paper is well organized and the proof of theorem is solid. The analysis of connectivity is good.

Weaknesses: Some analysis can be added.

---

> ### Author Response · Authors · 2022-08-02
> **Response**
>
> Thank you for reviewing our paper!
>
> 1. Weakness, the trade-off between robust accuracy and natural accuracy: The paper mainly focuses on the loss instead of the accuracy. However, these two metrics are highly correlated. When the loss is small, the accuracy is likely to be high. We will consider conducting some analysis on the accuracy part in our revision.
>
> 2. Similar to PGD and FSGM, auto-attack is another numerical way to calculate an attack. Since our theory studies the true attack, we expect to observe similar numerical results using auto-attack compared to PGD and FSGM. We will consider adding more experiments for different attacks.

---

### Author Response · Authors · 2022-08-02
**Common response to all reviewers**

We appreciate the reviewers spending time in reviewing our paper. Below is a clarification for our paper:

Reviewer rFja, Xy6A, T2si raised some questions about the phase transition phenomenon. We would like to emphasize that the transition in our paper is mathematical phase transition in an asymptotic sense, where the transition occurs within the range $\epsilon\asymp\epsilon^*$, instead of an exact critical value. The adversarial training trajectories under $\lim \epsilon/\epsilon^*=0$ and $\lim\epsilon/\epsilon^*=\infty$ are very different,  where we consider asymptotically $d$ and $n$ increase toward infinite. On the other hand,  under finite sample situation, as $\epsilon$ increases, the change is always smooth, but the rate of change shall be significantly different between attack range $\epsilon<\epsilon^*$ and  $\epsilon>\epsilon^*$.

1. This still counts as phase transition, where the boundary between phases is represented by asymptotic relationship. Similar terminology can be found in [1] (Reviewer rFja)

2. We run experiments for very large/small $\epsilon$ in Figures 3 and 4. (Reviewer Xy6A)

3. When $\epsilon$ just changes around $\epsilon^*$ with tiny perturbation, it is expected to only observe smooth changes in the adversarial robustness and related properties. (Reviewer T2si)

[1] Barbara M Smith. Constructing an asymptotic phase transition in random binary constraint satisfaction problems. Theoretical Computer Science, 265(1-2):265–283, 2001

---

### Meta-Review · Area_Chair_VKPN · 2022-08-31

**Recommendation:** Accept
**Confidence:** Less certain

**Metareview:**

Reviewers all agree that the theory in the paper is interesting and that it helps us understand the robustness accuracy tradeoff.   Several reviewers raise the issue that they are unsure about how “phase transition” is defined in this article, and whether the observed behavior is indeed a phase transition in a typical sense.  The reviewers also are unclear about why the values of epsilon were chosen for the experiments, and whether the experiments adequately demonstrate the behavior that the authors are intending to display.  I think the first issue is a semantic one, and does not rise to the level of rejecting the paper.  The second issue is not shared by all reviewers, and the justification for the choices of epsilon is explained in the rebuttal.  For this reason I feel that the outstanding issues have been adequately addressed by the authors.

Some suggestions to the authors for the camera ready: I feel that the main body of the paper is more fluid than the introduction, and I suggest carefully editing the introduction to ensure each sentence is clear.  I also suggest that the authors be clear about what is meant by "an asymptotic order" since some readers will not be clear on the meaning of this terminology.

**Award:**

No

---

### Decision · Program_Chairs · 2022-09-14

Accept